# LERD: Latent Event-Relational Dynamics for Neurodegenerative Classification

**Yicheng Feng** [* 1] **Hairong Chen** [* 2 3 4] **Ziyu Jia** [4 5 ✉] **Samir Bhatt** [1 6 ✉] **Hengguan Huang** [1 6 ✉]

## Abstract

Alzheimer's disease (AD) alters brain electrophysiology and disrupts multichannel EEG dynamics, making accurate and clinically useful EEG-based diagnosis increasingly important for screening and disease monitoring. However, many existing approaches rely on black-box classifiers and do not explicitly model the latent event timing and cross-channel coordination behind their decisions. To address these limitations, we propose LERD, an end-to-end Bayesian latent event–relational dynamical system that infers latent neural events and their relational structure directly from multichannel EEG without event or interaction annotations. LERD combines a continuous-time event inference module with a stochastic event-generation process to capture flexible temporal patterns, while incorporating an electrophysiology-inspired dynamical prior to guide learning in a principled way. We further provide theoretical analysis that yields a tractable IVP-based KL regularizer and stability guarantees for the inferred relational dynamics. Extensive experiments on synthetic benchmarks and two real-world AD EEG cohorts demonstrate that LERD consistently outperforms strong baselines and yields physiology-aligned rate, timing, and graph summaries that help characterize group-level dynamical differences.

---
[*]Equal contribution [1]Section of Health Data Science & AI, Department of Public Health, University of Copenhagen, Copenhagen, Denmark [2]Institute of Information Science, Beijing Jiaotong University, Beijing, China [3]College of Computing and Data Science, Nanyang Technological University, Singapore [4]Beijing Key Laboratory of Brainnetome and Brain-Computer Interface, Institute of Automation, Chinese Academy of Sciences, Beijing, China [5]Key Laboratory of Social Computing and Cognitive Intelligence (Dalian University of Technology), Ministry of Education [6]MRC Centre for Global Infectious Disease Analysis, Department of Infectious Disease Epidemiology, School of Public Health, Faculty of Medicine, Imperial College London, London, United Kingdom. Correspondence to: Hengguan Huang <hengguan.huang@sund.ku.dk>, Ziyu Jia <jia.ziyu@outlook.com>, Samir Bhatt <samir.bhatt@sund.ku.dk>.

*Proceedings of the 43rd International Conference on Machine Learning*, Seoul, South Korea. PMLR 306, 2026. Copyright 2026 by the author(s).

## 1. Introduction

Alzheimer's disease (AD) is a progressive neurodegenerative disorder and a major cause of dementia worldwide, with enormous personal and societal costs. Electroencephalography (EEG) offers a non-invasive, low-cost and widely available window into AD-related brain dysfunction. Decades of work have established robust macroscopic signatures, most notably *oscillatory slowing*, increased delta/theta power and reduced alpha/beta power, alongside alterations in large-scale interactions and synchrony across cortical regions (Jeong, 2004; Dauwels et al., 2010; Babiloni et al., 2021). These findings have motivated a growing interest in using EEG not only for automated AD assessment but also for mechanistic insight into how neural activity and network dynamics are disrupted over the course of the disease.

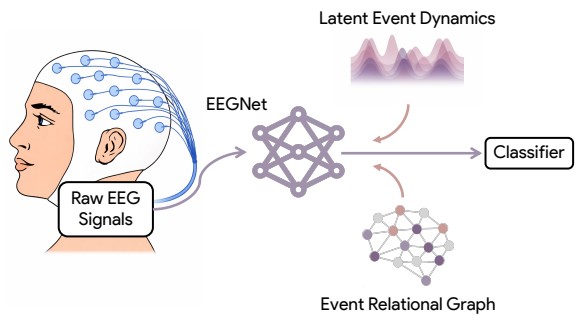

*Figure 1.* Overview of the LERD pipeline

Deep learning has substantially advanced EEG-based AD classification and staging. Convolutional and recurrent architectures, as well as graph- and transformer-based models, can ingest multichannel time series or time–frequency representations and achieve strong diagnostic performance (Ieracitano et al., 2020; Pineda et al., 2019; Vicchietti et al., 2023; Tawhid et al., 2025). Yet, most existing pipelines are designed as *label predictors* optimized primarily for accuracy rather than latent-event recovery from hand-crafted or learned features. They map rich spatiotemporal signals to labels, but typically do not model how *latent neural*

---
[1]Our code is available at: `https://github.com/hairongChenDavid/LERD`.

*events* and *interaction patterns* unfold and interact in time (Ehteshamzad, 2024; Acharya et al., 2025; Wang et al., 2024; Klepl et al., 2024). As a result, it remains difficult to relate model predictions back to dynamical structures that neuroscientists and clinicians can compare against established EEG phenomena, such as event-driven activity patterns, timing-derived interaction summaries and their disease-related distortions.

Building such a unifying, electrophysiology-aware dynamical framework is challenging for at least two reasons. First, scalp EEG is a noisy, frequency-dependent *linear mixture* of mesoscopic sources; reconstructing latent population activity from sensor-level data is an ill-posed inverse problem highly sensitive to modeling assumptions and head conductivity estimates (Michel & Brunet, 2019; Michel et al., 2004; Ma et al., 2026). Second, widely used interaction metrics suffer from *interpretational pitfalls*; volume conduction, common input and signal-to-noise differences can all induce spurious or inconsistent connectivity estimates across analysis pipelines unless dynamics and biophysical constraints are handled explicitly (Bastos & Schoffelen, 2016; Mahjoory et al., 2017). Together, these issues make it nontrivial to infer meaningful latent dynamics and directed graphs directly from raw or minimally processed EEG.

These observations raise a set of fundamental questions for EEG-based modeling of AD. *First*, can we design a neural dynamical system that infers latent, event-based activity trajectories directly from multichannel EEG, while respecting basic electrophysiological constraints such as plausible firing rates, refractory behavior and frequency ranges? *Second*, can we recover a directed, event-level interaction graph that captures how events in one channel probabilistically influence future events in others, in a way that is robust to mixing and noise and does not require spike or edge annotations? *Third*, can such a model simultaneously yield latent dynamical structure that supports scientific analysis and competitive performance on practical tasks, such as characterizing AD-related changes in network dynamics and improving EEG-based AD assessment?

We address these questions with **LERD**, a *Bayesian latent event–relational dynamical system* that infers event-driven latent dynamics and a conditional interaction graph directly from EEG, as shown in Figure 1. The key novelty is joint recovery of events, stochastic intervals, and event-lag graphs under one variational objective. At its core, LERD represents per-channel activity with an **Event Posterior Differential Equation (EPDE)**, whose solution provides a continuous-time representation of latent dynamics and expected next-event times. These expectations parameterize a **Mean–Evolving Lognormal Process (MELP)** that samples inter-event intervals from a log-normal mixture via reparameterized sampling, enabling flexible yet

tractable modeling of event statistics. To encode electrophysiological knowledge, we introduce a differentiable **leaky–integrate–and–fire (dLIF) prior** that imposes leak, refractory and rate constraints as well as plausible frequency ranges on the latent process. Finally, LERD infers a directed **event–relational graph (ERG)** by mapping cross-channel event lags through a smooth nonlinearity into edge weights, thereby summarizing how events in one channel precede, follow, or align with events in others. The entire model is trained end-to-end under a variational inference framework, for which we derive a tractable initial value problem (IVP)-based representation and finite-horizon surrogate for the event–prior KL under the dLIF rate, and we establish stability guarantees for the inferred ERG with respect to lag noise.

Importantly, our use of electrophysiology-inspired dLIF prior is *phenomenological* rather than a claim that scalp EEG directly reflects single-neuron dynamics. Scalp EEG has high temporal resolution (millisecond scale) but limited spatial specificity; we therefore treat the dLIF component as a structured prior on *population-level* latent rate dynamics (e.g., leak/refractory constraints), used to regularize an otherwise flexible neural dynamical system.

**Major Contributions.**

- We formulate annotation-free latent event–relation discovery and propose **LERD**, a unified Bayesian neural dynamical system that infers latent event processes and event–relational graph dynamics directly from multichannel EEG, without requiring spike or edge annotations.

- We incorporate an electrophysiology-informed **dLIF prior** into training, providing biophysically meaningful constraints on rates, refractoriness and frequency content of the inferred latent dynamics.

- We develop theory establishing a computationally tractable IVP-based event-prior KL representation and finite-horizon surrogate for the variational learning objective and a stability bound for LERD's inferred event–relational graph dynamics.

- We provide empirical evidence that LERD (i) accurately recovers latent event and graph dynamics that shed light on AD-related alterations in neural activity and interactions, and (ii) achieves superior performance over strong baselines on both synthetic benchmarks and real AD EEG datasets.

## 2. Related Work

A related line of research can be broadly viewed as *mechanistic Bayesian reasoning*: methods that move beyond

black-box prediction by discovering or explicitly modeling latent interactions and dynamics for understanding biological mechanisms from biomedical and biological data. Dynamic causal modeling is a canonical Bayesian example in neuroscience, using generative inversion to infer neurobiologically interpretable interactions from neuroimaging and electrophysiological measurements (Friston et al., 2003; Kiebel et al., 2009). Recent Bayesian deep models extend this spirit to latent microbiome interaction graphs, continuous-time adaptation under uncertainty, stochastic event-boundary dynamics, latent relational graph processes, and recurrent point-process modeling (Song et al., 2026; Huang et al., 2022; 2021; 2020; 2019). LERD follows this direction in AD EEG but targets a distinct annotation-free regime: both event times and event-lag relations are hidden, so the model must jointly infer population-level latent events and timing-derived interactions from multichannel signals. We next situate this contribution relative to two closest literatures: supervised EEG-based AD diagnosis and EEG dynamics/latent structure modeling.

## 2.1. EEG-based AD diagnosis with supervised pipelines

A large body of work frames EEG-based analysis of Alzheimer's disease (AD) as a supervised classification problem, where hand-crafted spectral or connectivity features (e.g., band power, topographies, and simple graph summaries) are extracted from multichannel recordings and then fed into conventional classifiers (Ieracitano et al., 2020). Recent studies further benchmark alternative feature–classifier combinations and examine protocol choices such as task versus resting-state recordings (Vicchietti et al., 2023; Kim et al., 2024). While these pipelines can achieve strong diagnostic accuracy, they typically summarize EEG into static descriptors and do not explicitly model the latent, event-driven dynamics and directed interactions that generate the signals. More recent studies continue to refine supervised AD–EEG pipelines with stronger models and evaluation protocols (Vicchietti et al., 2023; Kim et al., 2024; Tawhid et al., 2025). LERD is complementary: it preserves label prediction but makes latent event timing and event-lag relations explicit model outputs.

## 2.2. EEG dynamics and latent structure modeling

A complementary line of work seeks to model EEG dynamics more directly from a dynamical-systems perspective, emphasizing rapidly switching, whole-brain patterns and their temporal organization. EEG microstate analysis characterizes ongoing activity as sequences of quasi-stable scalp topographies (typically ∼60–120 ms) and provides an interpretable window into transient large-scale network dynamics (Michel & Koenig, 2018). Complementarily, Hidden Markov Modelling has been used to infer fast transient EEG network states and their transition structure (Hunyadi

et al., 2019).

Despite this progress, many dynamical approaches still operate on continuous-valued latent trajectories and do not explicitly recover directed, event-level interactions that explain how activity in one region influences future dynamics in others. Generative biophysical models such as dynamic causal modeling (DCM) provide a principled route to infer effective connectivity from EEG/MEG, but they typically require specifying an explicit neural mass model and a priori network structure (Kiebel et al., 2009). As a result, bridging transient EEG state dynamics with directed, event-driven interaction structure remains an open challenge.

Classical point-process state-space or GLM models provide powerful tools when event sequences are observed, for example in neural spike-train analysis (Smith & Brown, 2003; Truccolo et al., 2005). They are not direct drop-in baselines in our setting because the event times are latent and unannotated; using them would require an external event detector or additional supervision, which would change the problem definition. This is precisely the regime targeted by LERD.

## 3. Problem Formulation

We study unsupervised latent–event and relation discovery in multichannel time series with sequence-level labels. Given a dataset $\mathcal{D} = \{(X^{(n)}, Y^{(n)})\}_{n=1}^{N}$ with $X^{(n)} = \{x_c^{(n)}\}_{c=1}^{C}$ (e.g., multichannel EEG) and labels $Y^{(n)} \in \mathcal{Y}$ (e.g., AD vs. control), and *no* supervision on per-channel events or inter-channel relations, the objective is to infer (i) channel-wise latent event dynamics and (ii) a (possibly time-varying) relational/graphical structure among channels. For each channel $c \in \{1, \dots, C\}$ and sequence $n$, let $T_c^{(n)} = \{t_{c,k}^{(n)}\}_{k=1}^{K_c^{(n)}}$ denote the (unknown) latent event times, where $t_{c,k}^{(n)}$ is the $k$-th event time on channel $c$ in sequence $n$, and $K_c^{(n)}$ is the (latent) number of events on that channel. Let $\mathbf{p}^{(n)}(t) = \{p_c^{(n)}(t)\}_{c=1}^{C}$ denote the corresponding posterior event-time distributions. The relational structure is represented as a graph process $G^{(n)}(t)$ with adjacency $A^{(n)}(t) \in \mathbb{R}^{C \times C}$. We aim to recover $\{\mathbf{p}^{(n)}(t)\}_{n=1}^{N}$ and the conditional graph dynamics $P(G^{(n)}(\cdot) \mid \mathbf{p}^{(n)}(\cdot))$ from $\{X^{(n)}\}_{n=1}^{N}$, while using $(\mathbf{p}^{(n)}(\cdot), G^{(n)}(\cdot))$ as inputs to a downstream predictor for $Y^{(n)}$; importantly, $Y^{(n)}$ does not supervise the latent events or relations directly.

The interaction strength between channels is modeled as a function of the temporal *co-occurrence and ordering* of inferred events. Here an "event" denotes a population-level latent transition time inferred from scalp EEG, not a single-neuron spike. Accordingly, the ERG should be interpreted as a sensor-space, timing-derived latent interaction graph rather than source-level causal or DCM-style effective connectivity. This assumption is inspired by neurobiological

mechanisms of *spike-timing-dependent plasticity* (STDP), where near-coincident pre- and post-synaptic spikes modulate synaptic efficacy, while we do not claim that EEG-level edges identify synaptic plasticity or anatomical connectivity (Bi & Poo, 1998; Feldman, 2012).

# 4. Bayesian Neural Dynamical System

## 4.1. Overview

We introduce LERD, a Bayesian neural dynamical system for multichannel sequences that represents each channel with a latent *event* process and couples channels through a conditional *event–relational graph* (ERG) driven by the timing and ordering of inferred events. This design is motivated by settings where the clinically relevant signal resides in *when* events occur and *how* they align across channels. In EEG for Alzheimer's disease, for instance, oscillatory slowing and disrupted coordination suggest that event timing and cross-channel alignment are predictive, while neurobiological plasticity links near-coincident spikes to stronger coupling.

At a high level, LERD consists of three interacting components. First, an *event posterior differential equation* (EPDE) summarizes per-channel event dynamics by producing posterior distributions over latent event times $\{T_c^{(n)}\}$ given the observed multichannel sequence $X^{(n)}$. Second, a *mean–evolving lognormal mechanism* (MELP) uses EPDE outputs as mean parameters to generate stochastic inter-event timing between successive latent event times $t_{c,k}^{(n)}$, ensuring positive and flexible (potentially multimodal) timing statistics. Third, an *event–relational graph* $G^{(n)}(t)$ is inferred from event co-occurrence and cross-channel lags derived from $T^{(n)}$, via an STDP-shaped mapping that encodes how the timing of events on one channel is summarized as directed timing-dependent edge weights.

For each labeled sequence $(X^{(n)}, Y^{(n)})$, the triple $(X^{(n)}, T^{(n)}, G^{(n)})$ is passed to a decoder $p_\theta\big(Y^{(n)} \mid X^{(n)}, T^{(n)}, G^{(n)}\big)$ for downstream prediction (e.g., AD vs. control). Training is end-to-end via variational learning that jointly optimizes the EPDE and MELP while learning ERG dynamics under weak regularization; further details are given in the learning subsection.

## 4.2. Learning

We train LERD end-to-end through variational inference by *minimizing* a negative-ELBO-style objective. For labeled data $\{(X^{(n)}, Y^{(n)})\}_{n=1}^N$, let $T^{(n)}$ collect all latent event times $\{T_c^{(n)}\}_{c=1}^C$ and let $\tau^{(n)}$ collect the corresponding inter-event intervals $\{\tau_{c,k}^{(n)}\}$. The EPDE induces an approximate posterior $q_\phi\big(T^{(n)} \mid X^{(n)}\big)$, the MELP defines $q_\phi\big(\tau^{(n)} \mid X^{(n)}\big)$, and the decoder is $p_\theta\big(Y^{(n)} \mid X^{(n)}, T^{(n)}, G^{(n)}\big)$,

where $G^{(n)}$ is the ERG associated with $X^{(n)}$ and $\eta$ denotes ERG parameters.

Concretely, we minimize

$$
\mathcal{J}(\theta, \phi, \eta) = \sum_{n=1}^N \Big( -\mathbb{E}_{q_\phi}[\ell_n] + \mathrm{KL}_T^{(n)} + \mathrm{KL}_\tau^{(n)} \\
+ \beta\, \mathcal{R}_{\mathrm{ERG}}^{(n)} + \lambda_{\mathrm{LIF}}\, \mathcal{R}_{\mathrm{LIF}}^{(n)} \Big), \tag{1}
$$

where $\ell_n = \log p_\theta\big(Y^{(n)} \mid X^{(n)}, T^{(n)}, G^{(n)}\big)$ and $\mathrm{KL}_T^{(n)} := \mathrm{KL}\big(q_\phi(T^{(n)} \mid X^{(n)}) \,\|\, p_{\mathrm{dLIF}}(T)\big)$ compares the EPDE-induced path law to the electrophysiology-informed event prior $p_{\mathrm{dLIF}}(T)$, and $\mathrm{KL}_\tau^{(n)} := \mathrm{KL}\big(q_\phi(\tau^{(n)} \mid X^{(n)}) \,\|\, p_0(\tau)\big)$ penalizes deviation from a lognormal(-mixture) prior over inter-event intervals. The term $\mathcal{R}_{\mathrm{LIF}}^{(n)}$ softly enforces leaky–integrate–and–fire consistency on differentiable rate proxies read out from the EPDE state, with weight $\lambda_{\mathrm{LIF}} \geq 0$. The term $\mathcal{R}_{\mathrm{ERG}}^{(n)}$ is a weak, observable-based regularizer that nudges ERG edges toward experimental statistics computed from $X^{(n)}$ (e.g., correlation-based summaries), with strength $\beta \geq 0$.

**Challenges.** Three technical issues arise in optimizing (1). First, $\mathrm{KL}_T^{(n)}$ involves *path measures* induced by a differential equation and is intractable in closed form (it integrates over an infinite-dimensional trajectory); we therefore replace it with a tractable integral–rate surrogate that depends only on the dLIF rate $r(t)$, with a formal IVP representation in Theorem 4.1. Second, enforcing a LIF prior directly is difficult because the spike function in LIF is *non-differentiable*, which prevents straightforward use in gradient-based training; instead we introduce differentiable rate proxies and constrain them to follow dLIF laws via $\mathcal{R}_{\mathrm{LIF}}^{(n)}$. Third, ERG learning lacks ground-truth edges; to avoid over-constraining the graph, we only use the weak regularizer $\mathcal{R}_{\mathrm{ERG}}^{(n)}$ to bias edge strengths toward experimental observables, leaving the fine-grained graph structure to be driven by event lags inferred from the EPDE–MELP posterior.

Setting $\beta = 0$ removes this observable Fisher–$z$ anchoring, while setting $\lambda_{\mathrm{LIF}} = 0$ removes the dLIF prior. Table 4 reports no-prior, dLIF-only, ERG-only, and dual-prior variants, so LERD's gains are not obtained by forcing the ERG to reproduce Pearson correlations.

## 4.3. Prior: Electrophysiology–informed dLIF prior

We place a biophysical prior on latent event timing by instantiating each channel's latent events $T_c^{(n)} = \{t_{c,k}^{(n)}\}$ as a renewal process whose hazard is derived from a differentiable leaky–integrate–and–fire (dLIF) abstraction (Burkitt, 2006). For channel $c$, the (rescaled) membrane potential

evolves as

$$\frac{d}{dt} u_c(t) = b_c(t) - u_c(t), \qquad b_c(t) > 1, \qquad (2)$$

where $b_c(t)$ is an effective (learned) input drive. Given this membrane dynamics, the implied instantaneous *firing rate* is

$$r_c(t) = \Big[ -\log\big(1 - 1/b_c(t)\big) \Big]^{-1}. \qquad (3)$$

This rate induces a dLIF inter-event time density

$$p_{\mathrm{dLIF},c}(t) = r_c(t) \exp\Big( -\int_0^t r_c(s)\,ds \Big), \qquad (4)$$

and the resulting dLIF prior for channel $c$ is the renewal law $p_{\mathrm{dLIF}}(T_c^{(n)})$ with hazard $r_c(t)$. We parameterize $b_c(t)$ by a bounded neural mapping from learned embeddings, for example

$$b_c(t) = 1 + \mathrm{softplus}\big(g_\xi(z_c^{(n)}(t))\big), \qquad (5)$$

where $z_c^{(n)}(t)$ denotes features derived from $X^{(n)}$. This construction guarantees $b_c(t) > 1$ and thus $r_c(t) > 0$. Absolute and refractory effects are incorporated through a smooth gating factor $\alpha_c^{(n)}(t) \in (0, 1]$ constructed from recent events in $T_c^{(n)}$, using an effective rate $\widetilde{r}_c(t) = \alpha_c^{(n)}(t)\,r_c(t)$ to suppress implausible near–back–to–back spikes.

Because the hard spike nonlinearity is non–differentiable, we regularize *rates* rather than spikes. Concretely, the learning objective includes a dLIF consistency term

$$\mathcal{R}_{\mathrm{LIF}} = \sum_c \int_0^S \big(\widehat{r}_c^{(n)}(t) - r_c(t)\big)^2\,dt, \qquad (6)$$

where $\widehat{r}_c^{(n)}(t)$ is a differentiable rate proxy read from the EPDE state for sequence $n$ over a time horizon $[0, S]$. This encourages the learned rates to follow dLIF membrane dynamics without invoking non–differentiable spike functions (Neftci et al., 2019). The variational KL between the EPDE–induced path law $q_\phi\big(T_c^{(n)} \mid X^{(n)}\big)$ and $p_{\mathrm{dLIF}}(T_c^{(n)})$ is intractable in general; we therefore use the following tractable IVP representation, which depends on $r_c(t)$ and yields a stable finite-horizon surrogate for training while preserving the biophysical semantics of the prior.

**Theorem 4.1** (IVP representation of the event–prior KL under dLIF rates). *Let $q(t)$ be a strictly positive density on $[0, S]$ ($0 < S \leq \infty$), and assume the KL integrand below is integrable. Let $r : [0, S] \to [a, b] \subset (0, \infty)$ be measurable and define $R(t) = \int_0^t r(u)\,du$. On a finite EEG window, use the normalized dLIF first-event density*

$$p_r^S(t) = \frac{r(t) \exp[-R(t)]}{Z_S},$$

$$Z_S = \int_0^S r(v) \exp[-R(v)]\,dv = 1 - \exp[-R(S)], \qquad (7)$$

*with $Z_\infty = 1$. Let $h_S(t) = q(t)\log(q(t)/p_r^S(t))$, $m = -e^{-t}$ and $M(m) = -\log(-m)$. Define*

$$g(m) = -\frac{q(M(m))}{m} \log \frac{q(M(m))}{p_r^S(M(m))}, \qquad (8)$$

$$G'(m) = g(m), \qquad G(-1) = 0.$$

*For any finite horizon $S < \infty$,*

$$K_S := \mathrm{KL}(q\|p_r^S) = \int_0^S h_S(t)\,dt = G(-e^{-S}). \qquad (9)$$

*For $S = \infty$,*

$$K_\infty := \mathrm{KL}(q\|p_r^\infty) = \lim_{\varepsilon \downarrow 0} G(-\varepsilon),$$

$$|K_\infty - G(-\varepsilon)| \leq \int_{-\log \varepsilon}^\infty |h_\infty(t)|\,dt \to 0. \qquad (10)$$

The proofs are provided in the Appendix. In training, EEG windows have finite horizon, so the event-prior term is evaluated through the computable IVP integral in Eq. (9) using an ODE solver for Eq. (8). This term is used as a tractable prior-matching regularizer rather than an exact path-measure KL; its practical contribution is assessed by the prior ablations in Table 4.

We then analyze how entry–wise perturbations of lags affect the learned ERG when the edge map is exponential. For channels $i \neq j$ and time $t$, let the noise–free lag be $\Delta t_{ij}(t; T)$ and the perturbed lag be $\widetilde{\Delta t}_{ij}(t; T) = \Delta t_{ij}(t; T) + \xi_{ij}(t; T)$. Define the edge map $\phi_\alpha(x) = \exp(-\alpha|x|) \in [0, 1]$ with slope parameter $\alpha > 0$ and the (noise–free and perturbed) edges

$$e_{ij}(t; T) = \phi_\alpha(\Delta t_{ij}(t; T)),$$

$$\widetilde{e}_{ij}(t; T) = \phi_\alpha(\widetilde{\Delta t}_{ij}(t; T)).$$

The decoder uses the Monte–Carlo, time–averaged adjacencies

$$\bar{A}_{ij} = \frac{1}{MS} \sum_{m=1}^M \sum_{s=1}^S e_{ij}(t_m; T^{(s)}),$$

$$\widetilde{\bar{A}}_{ij} = \frac{1}{MS} \sum_{m=1}^M \sum_{s=1}^S \widetilde{e}_{ij}(t_m; T^{(s)}).$$

Proofs of the IVP representation and ERG stability results are provided in Appendix A.

### 4.4. Posterior: Event Posterior Differential Equation (EPDE)

For each sequence $(X^{(n)}, Y^{(n)})$ and channel $c$, let $q_c^{(n)}(t \mid x_c^{(n)})$ denote the density of the next event time given the observed channel signal $x_c^{(n)}$. If $\widetilde{t}_{c,k-1}^{(n)}$ denotes the previous

(predicted) event time on that channel, the expected next event time is

$$\tilde{t}_{c,k}^{(n)} = \int_{\tilde{t}_{c,k-1}^{(n)}}^{\infty} t \, q_c^{(n)}\big(t \mid x_c^{(n)}\big) \, dt. \qquad (11)$$

To express this update via an initial value problem (IVP), we introduce an auxiliary function $\Phi_c^{(n)}(t)$ whose derivative accumulates the contribution of $q_c^{(n)}$:

$$\big(\Phi_c^{(n)}\big)'(t) = -t \, q_c^{(n)}\big(t \mid x_c^{(n)}\big). \qquad (12)$$

Its initial value encodes the full expectation under $q_c^{(n)}$:

$$\Phi_c^{(n)}(0) = \int_0^{\infty} t \, q_c^{(n)}\big(t \mid x_c^{(n)}\big) \, dt. \qquad (13)$$

With this definition, the expected next event time in (11) can be written as the IVP solution evaluated at the previous event time:

$$\tilde{t}_{c,k}^{(n)} = \Phi_c^{(n)}\big(\tilde{t}_{c,k-1}^{(n)}\big). \qquad (14)$$

Directly solving (12)–14 and computing $\Phi_c^{(n)}(0)$ is intractable, so we approximate this mapping with a differentiable neural surrogate that updates the predicted next event time:

$$\tilde{t}_{c,k}^{(n)} = f_{\theta_\Phi}\big(\tilde{t}_{c,k-1}^{(n)}, x_c^{(n)}\big), \qquad (15)$$

implemented to ensure $\tilde{t}_{c,k}^{(n)} > \tilde{t}_{c,k-1}^{(n)}$, so that latent events remain strictly ordered in time.

Consequently, differentiating the ideal $\Phi_c^{(n)}(t)$ with respect to $t$ yields an *event-time posterior* of the form

$$q_c^{(n)}\big(t \mid x_c^{(n)}\big) = -\frac{\big(\Phi_c^{(n)}\big)'(t)}{t}, \qquad (16)$$

which we approximate with the EPDE parameterization. Across channels, the family $\{q_c^{(n)}(t \mid x_c^{(n)})\}_{c=1}^{C}$ provides a parametric approximation to the event-time posteriors $\{p_c^{(n)}(t)\}_{c=1}^{C}$ introduced in the problem formulation, and jointly defines the EPDE-induced posterior $q_\phi\big(T^{(n)} \mid X^{(n)}\big)$ over latent event times $T^{(n)} = \{T_c^{(n)}\}_{c=1}^{C}$ used in the variational objective. The predicted next-event times $\tilde{t}_{c,k}^{(n)}$ then serve as mean parameters for the mean–evolving lognormal mechanism (MELP) that models stochastic variability in event timing, as detailed in the next subsection.

### 4.5. Sampling: Mean–Evolving Lognormal Process (MELP)

For each sequence $X^{(n)}$ and channel $c$, given the previous event time $\tilde{t}_{c,i-1}^{(n)}$, the EPDE produces a $K$-dimensional vector of *candidate mean intervals* $\tilde{\boldsymbol{\tau}}_c^{(n)} =$

$\big(\tilde{\tau}_{c,1}^{(n)}, \ldots, \tilde{\tau}_{c,K}^{(n)}\big) \in \mathbb{R}_+^K$, together with mixture weights $\mathbf{w}_c^{(n)} = \big(w_{c,1}^{(n)}, \ldots, w_{c,K}^{(n)}\big) \in \Delta^{K-1}$ and scales $\mathbf{s}_c^{(n)} = \big(s_{c,1}^{(n)}, \ldots, s_{c,K}^{(n)}\big) \in \mathbb{R}_+^K$. MELP draws the inter–event interval $\tau_{c,i}^{(n)}$ from a lognormal mixture:

$$p\big(\tau_{c,i}^{(n)} \mid \cdot\big) = \sum_{j=1}^{K} w_{c,j}^{(n)} \operatorname{LogN}\big(\tau_{c,i}^{(n)}; \mu_{c,j}^{(n)}, (s_{c,j}^{(n)})^2\big),$$

$$(17)$$

where $\operatorname{LogN}(\cdot; \mu, s^2)$ denotes a lognormal density with log-mean $\mu$ and log-variance $s^2$.

For each component $j$, we choose $\mu_{c,j}^{(n)}$ as a function of $\tilde{\tau}_{c,j}^{(n)}$ and $s_{c,j}^{(n)}$ so that the *mean* of the corresponding lognormal distribution matches the EPDE-predicted interval $\tilde{\tau}_{c,j}^{(n)}$. This ties the mixture components' average inter-event times directly to the EPDE outputs, while the scales $s_{c,j}^{(n)}$ control uncertainty around these means.

Sampling from MELP is reparameterized to keep gradients pathwise:

$$k \sim \operatorname{Cat}\big(\mathbf{w}_c^{(n)}\big), \qquad \varepsilon \sim \mathcal{N}(0,1), \qquad (18)$$
$$\tau_{c,i}^{(n)} = \exp\big(\mu_{c,k}^{(n)} + s_{c,k}^{(n)} \varepsilon\big), \qquad t_{c,i}^{(n)} = t_{c,i-1}^{(n)} + \tau_{c,i}^{(n)}. \qquad (19)$$

During training we use a differentiable variant of (18) (e.g., Gumbel–Softmax) and take hard samples at test time. MELP guarantees positive inter-event intervals, captures multimodal timing statistics, and yields closed-form component-wise KL terms against a lognormal(-mixture) prior in the learning objective. Moreover, the mixture expectation $\mathbb{E}[\tau_{c,i}^{(n)}] = \sum_{j=1}^{K} w_{c,j}^{(n)} \tilde{\tau}_{c,j}^{(n)}$ is available in closed form and is used in downstream computations such as computing ERG lags.

Thus EPDE exposes a posterior over next-event times and MELP carries uncertainty and multi-modality in inter-event intervals, instead of reducing noisy EEG transitions to a deterministic boundary or an opaque hidden feature.

### 4.6. Event–relational graph (ERG)

We weakly bias $\bar{A}^{(n)}$ toward observable statistics computed from $X^{(n)}$ (e.g., Pearson correlations $s_{ij}^{(n)}$ between channels $i$ and $j$) via a simple Fisher–$z$ alignment. We map both the observed and ERG-implied correlations into $z$-space:

$$z_{ij}^{\operatorname{obs},(n)} = \operatorname{atanh}\big(s_{ij}^{(n)}\big),$$
$$z_{ij}^{\operatorname{pred},(n)} = \operatorname{atanh}\big(2\bar{A}_{ij}^{(n)} - 1\big), \qquad (20)$$

and define the ERG regularizer as

$$\mathcal{R}_{\operatorname{ERG}}^{(n)} = \sum_{i<j} \left[ \frac{\big(z_{ij}^{\operatorname{obs},(n)} - z_{ij}^{\operatorname{pred},(n)}\big)^2}{2\,\sigma^2} + \frac{1}{2}\log\sigma^2 \right],$$

$$(21)$$

where $\sigma > 0$ is a (fixed or globally learned) scale parameter controlling the strength of the alignment. This regularizer (weighted by $\beta$ in the learning objective) encourages consistency between ERG-implied connectivity and experimental statistics, while still allowing the detailed edge structure to be driven primarily by event lags inferred from the EPDE–MELP posterior. Pearson statistics are therefore weak observables, not graph labels: one-to-one agreement is neither expected nor desired, and an ERG that simply collapsed to Pearson correlation would add little beyond a standard correlation analysis. The implementation details and additional theoretical results are provided in Appendix F and A, respectively.

# 5. Experiments

In our experimental evaluations, we rigorously assessed the performance of LERD across synthetic benchmarks and real Alzheimer's disease (AD) EEG datasets. First, we validated LERD using synthetic event-sequence data, demonstrating its effectiveness in accurately inferring latent event dynamics compared to baseline neural ODE models. Subsequently, we conducted comprehensive experiments on two diverse EEG datasets (AD cohorts A and B), covering Alzheimer's disease, frontotemporal dementia, mild cognitive impairment, and healthy controls.

## 5.1. Toy Dataset

The synthetic benchmark uses three frequency bands, [5,10], [10,15], and [15,20] Hz. For each sequence we sample latent event rates from truncated normals centered in the corresponding band, draw event times from exponential inter-event intervals, and add controlled observation noise. This benchmark is included because, unlike real EEG, it provides known latent boundaries and rates, allowing direct structural recovery metrics such as boundary IoU rather than only downstream prediction accuracy.

The toy dataset experiments evaluated how well LERD and baseline neural ODE methods (NODE (Chen et al., 2018), ODE-RNN (Habiba & Pearlmutter, 2020), and STRODE (Huang et al., 2021)) recover latent event dynamics across distinct frequency bands ([5–10], [10–15], and [15–20] Hz). As summarized in Table 1, baseline methods achieved strong cosine similarity (CS) scores across all frequency bands, reflecting good sequence-level prediction performance. However, their intersection-over-union (IoU) scores were uniformly zero, indicating a fundamental limitation in capturing the latent event structure. This outcome highlights the common issue with purely data-driven neural approaches: despite excellent predictive accuracy, they often fail to recover the underlying generative mechanisms of data.

*Table 1.* Toy dataset results by frequency band. CS: Cosine Similarity, IoU: Intersection-over-Union.

| Freq Band (Hz) | Model | CS | Median Rate | 95% CI | IoU |
|---|---|---|---|---|---|
| [5, 10] | NODE | 0.951 | 1.000 | [1.000, 1.000] | 0.000 |
| [5, 10] | ODE-RNN | 0.951 | 1.000 | [1.000, 1.000] | 0.000 |
| [5, 10] | STRODE | 0.967 | 0.340 | [0.269, 0.410] | 0.000 |
| **[5, 10]** | **LERD (Ours)** | **0.983** | **7.532** | **[4.300, 14.867]** | **0.473** |
| [10, 15] | NODE | 0.951 | 1.000 | [1.000, 1.000] | 0.000 |
| [10, 15] | ODE-RNN | 0.951 | 1.000 | [1.000, 1.000] | 0.000 |
| [10, 15] | STRODE | 0.964 | 0.251 | [0.153, 0.348] | 0.000 |
| **[10, 15]** | **LERD (Ours)** | **0.982** | **12.503** | **[7.587, 24.918]** | **0.289** |
| [15, 20] | NODE | 0.951 | 1.000 | [1.000, 1.000] | 0.000 |
| [15, 20] | ODE-RNN | 0.951 | 1.000 | [1.000, 1.000] | 0.000 |
| [15, 20] | STRODE | 0.961 | 0.532 | [0.369, 0.695] | 0.000 |
| **[15, 20]** | **LERD (Ours)** | **0.976** | **18.843** | **[10.465, 35.244]** | **0.202** |

In contrast, LERD maintained similarly high CS scores while notably achieving meaningful IoU values (e.g., 0.473 at [5–10] Hz, 0.289 at [10–15] Hz, and 0.202 at [15–20] Hz). These non-zero IoU scores demonstrate that LERD captures latent structures consistent with the generative process and that its event-rate latents track the underlying oscillatory structure rather than acting as arbitrary embeddings. Moreover, the predicted event rates from LERD showed wider but informative uncertainty intervals, aligning closely with the true frequency band intervals. Such uncertainty quantification highlights LERD's capacity to provide both accurate predictions and useful latent summaries in noisy and ambiguous settings. Additional data generation protocols are provided in Appendix E.

Among the compared baselines, STRODE is the closest annotation-free timing model because it infers stochastic boundaries without event labels. NODE and ODE-RNN do not natively output latent event boundaries, and standard point-process/intensity models require observed event sequences or an external detector; this is why Table 1 treats their boundary IoU as a structural recovery diagnostic rather than as supervised event detection.

## 5.2. Alzheimer's Disease EEG Dataset

We extensively evaluated LERD on two diverse EEG datasets covering Alzheimer's disease (AD), frontotemporal dementia (FTD), mild cognitive impairment (MCI), and healthy controls. Across both datasets, LERD outperformed representative EEG classifiers and temporal/dynamical baselines such as CNNs, RNNs, and transformers. In Dataset AD cohort A, LERD showed a clear ability to distinguish AD, FTD, and healthy participants, with consistently lower performance variability across runs. This stability highlights its effectiveness in capturing subtle neural dynamics despite EEG heterogeneity. In the more unbalanced Dataset AD cohort B, where distinctions between moderate AD, MCI, and healthy controls are subtler, LERD still achieved the highest diagnostic accuracy. While some baselines showed significant performance fluctuations, LERD remained robust, balancing sensitivity and specificity—demonstrating resilience to noise and distribution shifts common in real-

world EEG data. Overall, LERD yields physiology-aligned latent summaries alongside strong predictive performance, supporting analyses of group-level spectral/connectivity differences under a strict cross-subject protocol. Additional dataset descriptions and baseline methods are presented in Appendix E.

*Table 2.* Results on Alzheimer's EEG datasets (mean ± std).

| Model | Dataset AD cohort A | | Dataset AD cohort B | |
|---|---|---|---|---|
| | Accuracy (%) | F1 (%) | Accuracy (%) | F1 (%) |
| EEGNet | 68.10 ± 7.10 | 66.49 ± 12.33 | 71.37 ± 26.02 | 60.85 ± 26.93 |
| LCADNet | 70.52 ± 5.09 | 68.12 ± 7.69 | 72.44 ± 24.16 | 49.38 ± 15.84 |
| LSTM | 70.52 ± 8.68 | 68.24 ± 9.59 | 77.89 ± 9.91 | 61.35 ± 20.35 |
| ATCNet | 64.71 ± 6.80 | 60.98 ± 4.55 | 71.09 ± 32.50 | 50.92 ± 23.61 |
| ADFormer | 69.35 ± 6.61 | 65.28 ± 7.87 | 82.38 ± 14.71 | 63.89 ± 12.09 |
| LEAD | 72.68 ± 4.57 | 69.98 ± 9.61 | 80.00 ± 15.55 | 62.21 ± 14.82 |
| **LERD** | **75.03 ± 8.29** | **72.69 ± 8.16** | **89.82 ± 8.39** | **64.87 ± 10.67** |

The high standard deviation of several Cohort B baselines should be read in light of the strict subject-level evaluation on a small and imbalanced cohort (notably only seven MCI subjects): a few subject-level majority-vote changes can produce large fold-to-fold swings. This is a baseline-specific stability issue under cross-subject aggregation rather than evidence of segment-level leakage.

*Table 3.* Per-window forward time on the AD EEG setting. LERD is slower than very small CNNs, but remains practical for offline clinical EEG analysis and is faster than the transformer/foundation-model baselines in this comparison.

| Model | Mean (ms) | Std (ms) |
|---|---|---|
| EEGNet | 0.713 | 0.013 |
| LCADNet | 0.875 | 0.063 |
| LSTM | 2.235 | 0.168 |
| ATCNet | 2.801 | 2.243 |
| LEAD | 11.148 | 0.115 |
| ADFormer | 19.247 | 3.411 |
| **LERD** | **9.209** | **1.139** |

### 5.3. Experimental Setup

We evaluate LERD under a five–fold cross–subject protocol to ensure that generalization is assessed on previously unseen participants rather than unseen windows from the same individuals. For each cohort, subjects are partitioned into five disjoint folds with stratification at the subject level so that the class proportions of Alzheimer's disease, frontotemporal dementia/mild cognitive impairment, and healthy controls are approximately preserved in every split. In each round, four folds are used for training, and one for testing; the roles of the folds are rotated until every fold serves exactly once in the held–out test set.

EEG is segmented into non–overlapping two–second windows (1,000 samples at 500 Hz) per subject and channel, followed by channel–wise $z$–normalization computed within the training portion of the active fold and then applied to validation and test windows of that fold.

*Table 4.* Ablation of spike-informed and connectome priors in **LERD** on Alzheimer's EEG Datasets. Accuracy and F1 (%) reported as mean ± std.

| Dataset | Variant | Accuracy (%) | F1 (%) |
|---|---|---|---|
| **AD cohort A** | No prior | 70.52 ± 11.83 | 65.46 ± 13.10 |
| | dLIF prior | 73.92 ± 9.84 | 71.41 ± 10.72 |
| | ERG prior | 72.75 ± 6.63 | 70.35 ± 7.91 |
| | Dual priors | **75.03 ± 8.29** | **72.69 ± 8.16** |
| **AD cohort B** | No prior | 83.22 ± 15.10 | 60.72 ± 12.37 |
| | dLIF prior | 87.98 ± 8.09 | 62.95 ± 9.24 |
| | ERG prior | 86.20 ± 9.96 | **65.63 ± 9.17** |
| | Dual priors | **89.82 ± 8.39** | 64.87 ± 10.67 |

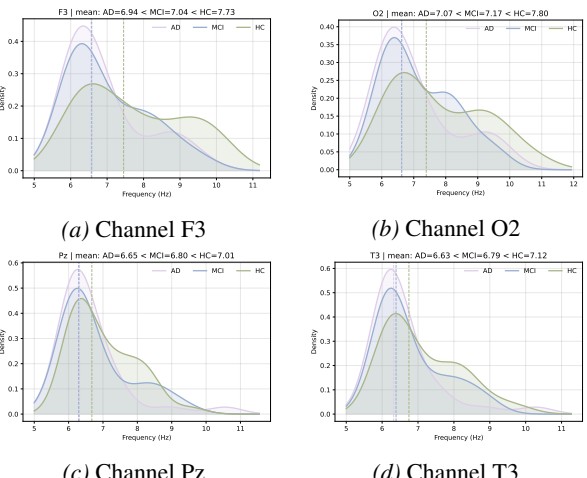

*(a)* Channel F3      *(b)* Channel O2

*(c)* Channel Pz      *(d)* Channel T3

*Figure 2.* Kernel density estimates of the inferred dLIF frequency distributions across Alzheimer's disease (AD), mild cognitive impairment (MCI), and healthy control (HC) groups for EEG channels F3, O2, Pz, and T3. The decreasing central frequency with increasing disease severity is consistent with established AD EEG slowing and serves as a model-derived latent summary.

Evaluation is conducted at subject levels. Subject–level predictions aggregate a subject's windows by majority vote over window–wise labels. We report accuracy, macro–F1, and summarize performance as the mean and standard deviation across the five test folds.

### 5.4. Ablation Studies

We ablated LERD to assess each prior's impact on predictive performance (Table 4). On AD Cohort A, the no-prior baseline is moderate; adding the dLIF prior boosts accuracy, while the ERG prior improves accuracy and F1 via inter-channel modeling. Combining both yields the best scores. On AD Cohort B, the pattern holds with larger gains: the dLIF prior gives the biggest accuracy lift under subtler classes, and the ERG prior raises F1. Their combination again performs best. Overall, each prior helps, and together they provide robust predictions on realistic EEG. One-factor sweeps in Appendix E.5 show that performance is not concentrated at a knife-edge setting of the auxiliary weights.

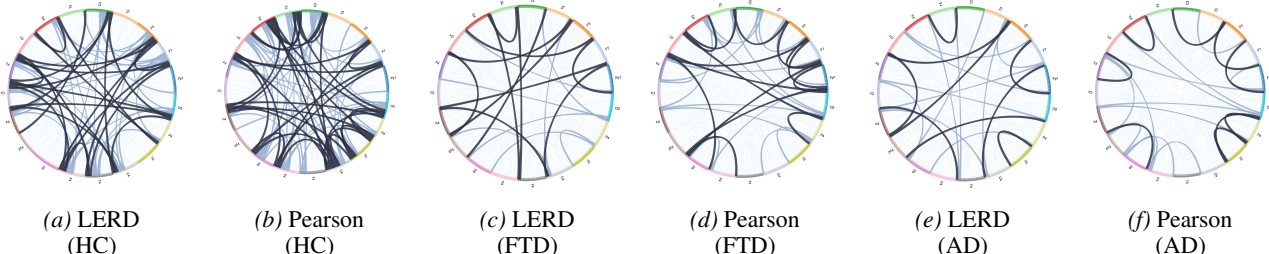

*(a)* LERD (HC)  *(b)* Pearson (HC)  *(c)* LERD (FTD)  *(d)* Pearson (FTD)  *(e)* LERD (AD)  *(f)* Pearson (AD)

*Figure 3.* Comparison of EEG connectivity graphs inferred by LERD versus Pearson correlation-based priors across healthy controls (HC), frontotemporal dementia (FTD), and Alzheimer's disease (AD) groups.

# 6. Visualizations

Our visualizations summarize three model-derived quantities: dLIF-constrained latent rates, timing-derived event-relational graphs, and synthetic boundary-time recovery with ground truth.

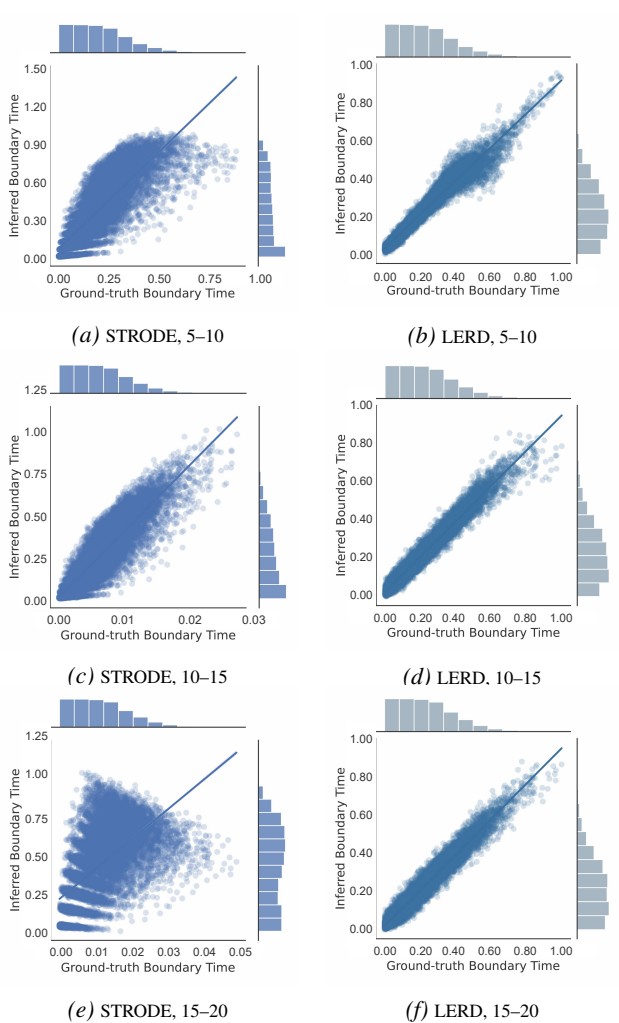

*(a)* STRODE, 5–10  *(b)* LERD, 5–10

*(c)* STRODE, 10–15  *(d)* LERD, 10–15

*(e)* STRODE, 15–20  *(f)* LERD, 15–20

*Figure 4.* Predicted vs. ground-truth boundary times across frequency bands: STRODE vs. LERD.

## 6.1. dLIF Inferred Frequency Visualization

Figure 2 shows inferred dLIF frequency distributions for representative frontal, occipital, parietal and temporal channels. The distributions follow the expected AD-related slowing trend: HC tends to have higher central latent frequency, MCI is intermediate, and AD shifts lower. The four plotted channels span distinct regions rather than selected effects; Appendix E.6 reports the remaining 19 channels, where HC is highest in 18/19 and higher than MCI in all 19.

## 6.2. Graph Connectivity Visualization

Figure 3 compares LERD's event-relational graphs (ERGs) with Pearson connectivity graphs. HC shows denser timing-derived connectivity, while FTD and AD exhibit sparser/weaker links, consistent with dementia EEG findings. Pearson similarity is expected only weakly because the Fisher–$z$ term regularizes but does not supervise the ERG. We interpret ERGs as sensor-level timing summaries, not source-resolved causal connectivity.

## 6.3. Boundary Time Prediction Visualization

Figure 4 visualizes predicted versus ground-truth synthetic boundary times. STRODE has native annotation-free boundary inference and is therefore the most meaningful visual comparator; NODE and ODE-RNN require ad hoc pseudo-boundaries and show near-zero IoU in Table 1. LERD stays closer to the diagonal across all bands, matching the quantitative boundary-recovery gains.

# 7. Conclusion

LERD is a Bayesian latent event–relational dynamical system for annotation-free recovery of population-level EEG events and timing-derived interaction graphs. By combining EPDE, MELP and an electrophysiology-inspired dLIF prior, LERD yields competitive AD classification together with rate, timing and graph summaries aligned with known neurodegenerative EEG phenomena. The theory provides a computable IVP event-prior KL representation and graph-stability guarantees, while experiments on synthetic and real EEG data demonstrate accurate latent recovery and strong diagnostic performance.

## Impact Statement

This paper presents work whose goal is to advance the field of machine learning for modeling and classifying neurodegenerative disease from EEG. If deployed responsibly, such models could support earlier screening and improve understanding of disease-related electrophysiological dynamics.

Potential negative impacts include misuse as a standalone diagnostic tool, leading to false positives/negatives and downstream harm; reduced performance under dataset shift (different sites, devices, demographics, or protocols); and privacy risks because EEG is sensitive health data. To mitigate these risks, we emphasize that LERD is intended as a research and decision-support model rather than a replacement for clinical judgment, and that any real-world use should include rigorous external validation, fairness and subgroup analyses, uncertainty reporting, and strong data governance (consent, de-identification, and secure storage).

## Acknowledgments

We thank all reviewers, SPC, and AC for their valuable comments. S.B. acknowledges support from the Novo Nordisk Foundation via The Novo Nordisk Young Investigator Award (NNF20OC0059309). S.B. acknowledges support from The Eric and Wendy Schmidt Fund For Strategic Innovation via the Schmidt Polymath Award (G-22-63345) which also supports HH and LM. S.B. acknowledges support from the Novo Nordisk Foundation via the Global Pathogen Analysis Platform (GPAP) (NNF26SA0109818). Z.J. acknowledges support from the Youth Science Fund Project of the National Natural Science Foundation of China (Grant No. 62306317), the Open Research Fund of the Key Laboratory of Social Computing and Cognitive Intelligence (Dalian University of Technology), Ministry of Education (Grant No. SCCI2025YB01), the Beijing Nova Program (Grant No. 20250484804), and the Young Elite Scientists Sponsorship Program of the Beijing High Innovation Plan (Grant No. 20250912).

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

## A. Theory

**Lemma A.1** (Shift–stability of IVPs (Huang et al., 2021)). *Let $e > 0$ and $U \subset \mathbb{R}^n$ be open. Let $f_1, f_2 : [a - 2e, a) \to \mathbb{R}^n$ be continuously differentiable with $\|f_1'\| \leq M$ for some $M > 0$. Consider*

$$y_1'(t) = f_1(t), \quad y_1(a - e) = x_1, \qquad y_2'(t) = f_2(t) = f_1(t - e), \quad y_2(a - e) = x_2.$$

*Then, as $e \to 0^+$,*

$$\lim_{e \to 0^+} \left( \lim_{t \nearrow a} \|y_1(t) - y_2(t)\| \right) \ \leq \ \lim_{e \to 0^+} \|x_1 - x_2\|.$$

**Theorem A.2** (Entry–wise and matrix stability). *The exponential edge map is globally $\alpha$–Lipschitz: for all $x, y \in \mathbb{R}$,*

$$\big| \phi_\alpha(x) - \phi_\alpha(y) \big| \leq \alpha \, |x - y|. \tag{22}$$

*Consequently, for any $(i, j, t, T)$,*

$$\big| \widetilde{e}_{ij}(t; T) - e_{ij}(t; T) \big| \ \leq \ \alpha \, |\xi_{ij}(t; T)|. \tag{23}$$

*Averaging over time and Monte–Carlo samples yields the entry–wise bound*

$$\big| \widetilde{A}_{ij} - \bar{A}_{ij} \big| \ \leq \ \alpha \, \overline{|\xi_{ij}|}, \qquad \overline{|\xi_{ij}|} \ := \ \frac{1}{MS} \sum_{m=1}^{M} \sum_{s=1}^{S} |\xi_{ij}(t_m; T^{(s)})|, \tag{24}$$

*and the matrix (Frobenius–norm) bound*

$$\|\widetilde{A} - \bar{A}\|_{\mathrm{F}} \ \leq \ \frac{\alpha}{MS} \sum_{m=1}^{M} \sum_{s=1}^{S} \|\Xi^{(m,s)}\|_{\mathrm{F}}, \qquad \Xi^{(m,s)} := \big[ \xi_{ij}(t_m; T^{(s)}) \big]_{i \neq j}, \tag{25}$$

*hence $\|\widetilde{A} - \bar{A}\|_{\mathrm{F}} \leq \alpha \, \overline{\|\Xi\|_{\mathrm{F}}}$ with $\overline{\|\Xi\|_{\mathrm{F}}}$ the average Frobenius norm of lag–noise matrices.*

**Corollary A.3** (Deterministic and Gaussian perturbation bounds). *(i) (Uniformly bounded noise). If $|\xi_{ij}(t; T)| \leq \varepsilon_\infty$ almost surely, then*

$$\big| \widetilde{A}_{ij} - \bar{A}_{ij} \big| \leq \alpha \, \varepsilon_\infty, \qquad \|\widetilde{A} - \bar{A}\|_{\mathrm{F}} \leq \alpha \, \varepsilon_\infty \, \sqrt{C(C - 1)}. \tag{26}$$

*(ii) (Gaussian timing noise). Suppose $\{\xi_{ij}(t_m; T^{(s)})\}_{m,s}$ are independent $\mathcal{N}(0, \sigma^2)$ variables, conditional on the sampled event paths. Let $\Delta_{ij} := \widetilde{A}_{ij} - \bar{A}_{ij}$. Then*

$$\mathbb{P}(|\Delta_{ij} - \mathbb{E}\Delta_{ij}| \geq \tau) \leq 2 \exp\left( -\frac{MS \, \tau^2}{2\alpha^2 \sigma^2} \right), \tag{27}$$

*and the bias and expected matrix perturbation obey*

$$|\mathbb{E}\Delta_{ij}| \leq \mathbb{E}\big[ |\Delta_{ij}| \big] \leq \alpha \, \sigma \sqrt{\frac{2}{\pi}}, \qquad \mathbb{E}\big[ \|\widetilde{A} - \bar{A}\|_{\mathrm{F}} \big] \leq \alpha \, \sigma \, \sqrt{C(C - 1)}. \tag{28}$$

**Implication.** Small perturbations in EPDE/MELP lags translate linearly (in $\alpha$) to entry–wise ERG changes. Under Gaussian timing noise, averaged edge perturbations concentrate around a bounded bias at the usual $1/\sqrt{MS}$ scale, with explicit constants controlled by the edge-map slope and noise magnitude.

## B. Proof of Theorem 4.1

*Proof.* First, the normalizing constant in Eq. (7) satisfies

$$Z_S = \int_0^S r(v) e^{-R(v)} \, dv = \int_0^S -\frac{d}{dv} e^{-R(v)} \, dv = 1 - e^{-R(S)}.$$

If $S = \infty$, the assumption $r(v) \geq a > 0$ implies $R(v) \to \infty$ and hence $Z_\infty = 1$. Thus $p_r^S$ is a valid first-event density on the observation horizon.

Let $h(t) = q(t) \log(q(t)/p_r^S(t))$. The change of variables $m = -e^{-t}$ gives $M(m) = -\log(-m)$ and $dt = -(1/m)\,dm$. Since $t = 0$ maps to $m = -1$ and $t = S$ maps to $m = -e^{-S}$, for any finite $S$,

$$\int_0^S h(t)\,dt = \int_{-1}^{-e^{-S}} h(M(m))\Big(-\frac{1}{m}\Big)\,dm = \int_{-1}^{-e^{-S}} g(m)\,dm = G(-e^{-S}),$$

which proves Eq. (9).

For $S = \infty$, the same finite-interval identity with $T_\varepsilon = -\log\varepsilon$ gives

$$G(-\varepsilon) = \int_0^{T_\varepsilon} h(t)\,dt.$$

Therefore,

$$|\text{KL}(q\|p_r^\infty) - G(-\varepsilon)| = \left|\int_{T_\varepsilon}^\infty h(t)\,dt\right| \le \int_{T_\varepsilon}^\infty |h(t)|\,dt,$$

and the right-hand side converges to zero by integrability of the KL integrand. This proves Eq. (10). □

## C. Proof of Theorem A.2

*Proof. (Lipschitzness).* The absolute value is 1–Lipschitz: $\big||x| - |y|\big| \le |x - y|$. The function $u \mapsto e^{-\alpha u}$ on $u \ge 0$ has derivative $|-\alpha e^{-\alpha u}| \le \alpha$, hence it is $\alpha$–Lipschitz on $\mathbb{R}_{\ge 0}$. By composition of Lipschitz maps,

$$|\phi_\alpha(x) - \phi_\alpha(y)| = \big|e^{-\alpha|x|} - e^{-\alpha|y|}\big| \le \alpha\,\big||x| - |y|\big| \le \alpha\,|x - y|,$$

establishing Eq. (22). Taking $y = x + \xi$ gives Eq. (23).

*(Averaging).* Using linearity of the average and triangle inequality,

$$\big|\widetilde{\bar{A}}_{ij} - \bar{A}_{ij}\big| = \left|\frac{1}{MS}\sum_{m,s}\big(\widetilde{e}_{ij}(t_m; T^{(s)}) - e_{ij}(t_m; T^{(s)})\big)\right| \le \frac{1}{MS}\sum_{m,s}|\widetilde{e}_{ij} - e_{ij}| \le \frac{\alpha}{MS}\sum_{m,s}|\xi_{ij}(t_m; T^{(s)})|,$$

which is Eq. (24).

*(Matrix bound).* Define $\Delta^{(m,s)} := [\widetilde{e}_{ij}(t_m; T^{(s)}) - e_{ij}(t_m; T^{(s)})]_{i \ne j}$, so $\widetilde{\bar{A}} - \bar{A} = \frac{1}{MS}\sum_{m,s}\Delta^{(m,s)}$. By the triangle inequality of the Frobenius norm,

$$\|\widetilde{\bar{A}} - \bar{A}\|_{\text{F}} \le \frac{1}{MS}\sum_{m,s}\|\Delta^{(m,s)}\|_{\text{F}}.$$

Entry–wise inequality Eq. 23 implies $\|\Delta^{(m,s)}\|_{\text{F}} \le \alpha\,\|\Xi^{(m,s)}\|_{\text{F}}$, giving Eq. (25). □

## D. Proof of Corollary A.3

*Proof.* (i) From Eq. (24), $\overline{|\xi_{ij}|} \le \varepsilon_\infty$, giving the entry–wise claim. For the matrix bound, $\|\Xi^{(m,s)}\|_{\text{F}} \le \varepsilon_\infty\sqrt{C(C-1)}$ for every $(m, s)$; applying Eq. (25) gives Eq. (26).

(ii) Fix $(i, j)$ and condition on the unperturbed event lags. Let $N = MS$ and write

$$F(\xi_1, \ldots, \xi_N) = \frac{1}{N}\sum_{n=1}^N \Big[\phi_\alpha(x_n + \xi_n) - \phi_\alpha(x_n)\Big],$$

where $\{x_n\}$ denote the corresponding noise-free lags. By Theorem A.2, $F$ is $\alpha/\sqrt{N}$-Lipschitz with respect to the Euclidean norm. Gaussian concentration for Lipschitz functions gives

$$\mathbb{P}\big(|F - \mathbb{E}F| \ge \tau\big) \le 2\exp\Big(-\frac{N\tau^2}{2\alpha^2\sigma^2}\Big),$$

which is Eq. (27) since $F = \Delta_{ij}$.

For the bias, Eq. (23) gives

$$|\Delta_{ij}| \leq \frac{\alpha}{N} \sum_{n=1}^{N} |\xi_n|.$$

Taking expectations and using $\mathbb{E}|\xi| = \sigma\sqrt{2/\pi}$ for $\xi \sim \mathcal{N}(0, \sigma^2)$ yields the entry–wise bounds in Eq. (28). For the Frobenius term, Jensen's inequality gives $\mathbb{E}\|\widetilde{A} - \bar{A}\|_{\mathrm{F}} \leq (\sum_{i \neq j} \mathbb{E}\Delta_{ij}^2)^{1/2}$, and Eq. (23) implies $\mathbb{E}\Delta_{ij}^2 \leq \alpha^2\sigma^2$, yielding the stated matrix bound. $\qquad\square$

## E. Additional Experimental Details

### E.1. Alzheimer's Disease EEG Baseline Methods

To rigorously evaluate our proposed method, we benchmarked it against several representative deep learning approaches commonly utilized for EEG analysis. These baselines include convolutional, recurrent, attention-based, and transformer-based models, each demonstrating distinct strengths for capturing various aspects of EEG signal patterns.

**EEGNet** (Lawhern et al., 2018) is a compact convolutional neural network initially developed for EEG-based brain–computer interfaces. It integrates depthwise and separable convolutions to effectively capture temporal, spatial, and frequency-specific characteristics in EEG data, making it a well-recognized lightweight yet powerful model in EEG classification tasks.

**LCADNet** (Kachare et al., 2024) is specifically tailored for Alzheimer's disease detection from EEG data. Utilizing optimized convolutional structures designed for computational efficiency without sacrificing discriminative power, LCADNet achieves competitive performance in resource-limited environments, making it a strong baseline for EEG-based AD diagnosis.

**LSTM** (Zhang & Yao, 2021) embodies recurrent neural networks tailored for modeling temporal dependencies inherent in EEG signals. By maintaining and updating hidden states across sequences, LSTMs effectively capture long-term dynamics and temporal correlations, making them naturally suitable for sequential EEG analyses.

**ATCNet** (Altaheri et al., 2023) employs a physics-informed architecture combining temporal convolutions with attention mechanisms. Originally proposed for motor imagery EEG classification, it effectively captures both local temporal details and global dependencies, showcasing adaptability across various EEG applications.

**ADformer** (Wang et al., 2024) is a multi-granularity transformer specifically crafted for Alzheimer's disease evaluation using EEG signals. It utilizes multi-scale attention mechanisms to concurrently model fine-grained and coarse-grained temporal information, setting a high-performance standard in EEG-based AD diagnostics.

**LEAD** (Wang et al., 2025) exemplifies the recent advancement toward large-scale foundation models in EEG analysis. Pre-trained extensively on vast EEG datasets and fine-tuned for Alzheimer's disease detection, LEAD leverages transfer learning to provide robust, generalizable EEG representations, establishing a new benchmark in EEG-based clinical assessments.

### E.2. Dataset Descriptions

#### E.2.1. Toy Dataset and Data Generation

**Frequency bands and sampling.** To systematically evaluate our model's ability to capture latent event dynamics, we constructed synthetic datasets with clearly defined frequency bands. We generated latent event rates $\lambda$ from truncated normal distributions centered at the midpoint of each target frequency band: low band [5–10 Hz] with $\lambda \sim \mathrm{TruncNormal}(\mu = 7.5, \sigma = 1.0; [5, 10])$, middle band [10–15 Hz] with $\lambda \sim \mathrm{TruncNormal}(\mu = 12.5, \sigma = 1.0; [10, 15])$, and high band [15–20 Hz] with $\lambda \sim \mathrm{TruncNormal}(\mu = 17.5, \sigma = 1.0; [15, 20])$. This design ensures concentrated event-rate distributions within each band while avoiding frequencies outside the desired range.

**Data scale and splitting strategy.** For each frequency band, we independently generated three data splits: a training set with 150 distinct event rates, each having 50 sequences (7,500 sequences total); a validation set with 25 new event rates and 50 sequences per rate (1,250 sequences total); and a test set with an additional 25 new event rates, again with 50 sequences per rate (1,250 sequences total). Importantly, no overlap of event rates occurs across training, validation, and test splits to ensure proper evaluation of model generalization.

**Sequence generation.** Each synthetic sequence comprises 20 observations, constructed by sampling inter-event times $\Delta t_i$ from an exponential distribution with parameter $\lambda$. The event timestamps $t_i$ are obtained cumulatively by $t_i = \sum_{j=1}^{i} \Delta t_j$. Observations $y_i$ are subsequently generated using the relationship:

$$y_i = \sin(t_i) + \eta_i, \quad \eta_i \sim \mathcal{N}(0, \sigma_\eta^2),$$

where the default noise level is $\sigma_\eta = 0.07$. Additional sensitivity analyses varied $\sigma_\eta$ within $\{0.05, 0.10, 0.15\}$ to assess model robustness.

**Evaluation methodology.** Model performance was comprehensively evaluated using three criteria: (1) classification score (CS) assessing sequence-level predictive accuracy, (2) uncertainty calibration, quantified through the median and 95% confidence interval of the estimated event rate $\hat{\lambda}$, obtained via nonparametric resampling within each frequency band, and (3) structural fidelity, measured using intersection-over-union (IoU) between the predicted latent structure and the ground-truth event patterns.

### E.2.2. ALZHEIMER'S DISEASE EEG DATASET

**Dataset AD Cohort A** (Miltiadous et al., 2024) consists of resting-state, eyes-closed EEG recordings from a total of 88 participants, categorized into 36 individuals diagnosed with Alzheimer's disease (AD), 23 patients with frontotemporal dementia (FTD), and 29 healthy control subjects (HC). The EEG data were collected using 19 electrodes arranged according to the international 10–20 placement system. The recordings have a sampling rate of 500 Hz and an average duration ranging from approximately 12 to 14 minutes per subject. Provided in adherence to the Brain Imaging Data Structure (BIDS) standard, the dataset includes both raw and preprocessed EEG signals, enabling robust comparative analysis across different dementia subtypes.

**Dataset AD Cohort B** (Sadegh-Zadeh et al., 2023) includes resting-state EEG data from 168 participants, segmented into 59 moderate Alzheimer's disease patients (AD), 7 individuals diagnosed with mild cognitive impairment (MCI), and 102 healthy controls (HC). EEG recordings were acquired using the standardized 10–20 electrode placement system, with data presented in MATLAB (.mat) format. Accompanying the EEG data are Mini-Mental State Examination (MMSE) scores, providing cognitive assessments for participants. This dataset is particularly tailored for the distinction of AD from MCI, thus serving as a valuable resource for investigations aimed at early Alzheimer's disease diagnosis.

### E.3. Computational cost

To contextualize runtime/complexity, we report a per-window estimate of floating point operations (FLOPs) and parameter counts for a single 2-s EEG window (19 channels, 500 Hz). While LERD is heavier than lightweight CNN backbones (e.g., EEGNet), it remains substantially smaller than transformer-based baselines.

*Table 5.* Approximate computational cost for one 2-s EEG window (19 channels, 500 Hz).

| Model | FLOPs | Params |
|-------|-------|--------|
| **LERD** | 60.176M | 3.103M |
| EEGNet | 9.989M | 1.232K |
| LCADNet | 7.783M | 151.302K |
| LSTM | 114.82M | 5.5K |
| ATCNet | 328.74M | 4.3K |
| ADFormer | 3.636G | 5.34M |
| LEAD | 0.766G | 3.33M |

### E.4. Additional evaluation metrics

For completeness, we report sensitivity, specificity, and AUC (mean $\pm$ s.d. across the five cross-subject folds) for both cohorts.

*Table 6.* Sensitivity, specificity, and AUC on Dataset AD cohort A.

| Model | Sensitivity | Specificity | AUC |
|-------|-------------|-------------|-----|
| EEGNet | $0.6830 \pm 0.1050$ | $0.8463 \pm 0.0355$ | $0.7646 \pm 0.0695$ |
| ADFormer | $0.7017 \pm 0.0833$ | $0.8495 \pm 0.0366$ | $0.7756 \pm 0.0597$ |
| LSTM | $0.6756 \pm 0.0692$ | $0.8426 \pm 0.0382$ | $0.7591 \pm 0.0529$ |
| ATCNet | $0.6844 \pm 0.0627$ | $0.8350 \pm 0.0334$ | $0.7597 \pm 0.0460$ |
| LEAD | $0.7067 \pm 0.0498$ | $0.8594 \pm 0.0216$ | $0.7830 \pm 0.0343$ |
| LCADNet | $0.7006 \pm 0.0491$ | $0.8445 \pm 0.0358$ | $\mathbf{0.8462 \pm 0.0495}$ |
| **LERD** | $\mathbf{0.7493 \pm 0.0841}$ | $\mathbf{0.8793 \pm 0.0377}$ | $0.8143 \pm 0.0603$ |

*Table 7.* Sensitivity, specificity, and AUC on Dataset AD cohort B.

| Model | Sensitivity | Specificity | AUC |
|-------|-------------|-------------|-----|
| EEGNet | $0.5544 \pm 0.0801$ | $0.8717 \pm 0.0921$ | $0.9494 \pm 0.0445$ |
| ADFormer | $0.6249 \pm 0.0430$ | $0.9546 \pm 0.0306$ | $\mathbf{0.9714 \pm 0.0381}$ |
| LSTM | $0.5167 \pm 0.0867$ | $0.9038 \pm 0.0387$ | $0.9486 \pm 0.0314$ |
| ATCNet | $0.6412 \pm 0.0313$ | $\mathbf{0.9628 \pm 0.0082}$ | $0.9650 \pm 0.0366$ |
| LEAD | $0.5827 \pm 0.1291$ | $0.9224 \pm 0.0484$ | $0.9568 \pm 0.0311$ |
| LCADNet | $0.6427 \pm 0.1045$ | $0.9500 \pm 0.0257$ | $0.9657 \pm 0.0344$ |
| **LERD** | $\mathbf{0.6891 \pm 0.1182}$ | $0.9500 \pm 0.0388$ | $0.8383 \pm 0.0578$ |

## E.5. Auxiliary-weight sensitivity

To check whether LERD depends on a knife-edge auxiliary-weight choice, we performed one-factor sensitivity sweeps on Cohort A while keeping the remaining training protocol fixed. Performance is strongest in a moderate range for each regularizer and degrades mainly when weights become poorly scaled.

*Table 8.* One-factor sensitivity sweeps for auxiliary regularization weights on AD cohort A. Accuracy is reported in percentage.

| dLIF weight | Acc. | ERG weight | Acc. | IVP-KL weight | Acc. |
|-------------|------|------------|------|---------------|------|
| 1 | 71.70 | 1 | 70.39 | 0 | 71.63 |
| $10^{-1}$ | 75.03 | $10^{-1}$ | 70.52 | $10^{-10}$ | 72.81 |
| $10^{-2}$ | 73.92 | $10^{-2}$ | 73.86 | $5 \times 10^{-10}$ | 75.03 |
| $10^{-3}$ | 69.35 | $10^{-3}$ | 66.99 | $2 \times 10^{-9}$ | 73.86 |
| $10^{-4}$ | 68.24 | $10^{-4}$ | 63.73 | $10^{-9}$ | 72.61 |
| $10^{-5}$ | 72.81 | $10^{-5}$ | 72.75 | $10^{-8}$ | 70.52 |
| $10^{-6}$ | 70.46 | $10^{-6}$ | 73.92 | $10^{-7}$ | 70.46 |
| $10^{-7}$ | 67.06 | $10^{-7}$ | 69.22 | $10^{-6}$ | 71.57 |
| $10^{-8}$ | 72.81 | $10^{-8}$ | 75.03 | $10^{-5}$ | 70.59 |
| $10^{-9}$ | 72.81 | $10^{-9}$ | 72.75 | $10^{-4}$ | 68.30 |
| $10^{-10}$ | 71.63 | $10^{-10}$ | 70.52 | $10^{-3}$ | 69.41 |
| 0 | 73.92 | 0 | 72.75 | | |

## E.6. Additional dLIF frequency channels

Table 9 reports the non-plotted channels used to check whether the frequency-slowing trend in Figure 2 is representative. Across these 19 channels, HC has the highest inferred central frequency in 18/19 and is higher than MCI in all 19.

*Table 9.* Additional inferred central frequencies (Hz) by group for all 19 channels.

| Channel | HC | MCI | AD | Channel | HC | MCI | AD |
|---------|------|------|------|---------|------|------|------|
| T5 | 9.8183 | 8.7469 | 9.2903 | T4 | 8.7062 | 8.1207 | 8.9866 |
| C4 | 9.5672 | 8.5863 | 9.3575 | P4 | 8.8150 | 8.3549 | 7.9066 |
| Fp1 | 9.6450 | 8.7443 | 9.0170 | F4 | 8.5122 | 7.5947 | 7.6261 |
| C3 | 9.4277 | 8.5671 | 9.2716 | T6 | 8.3738 | 7.5920 | 7.6467 |
| Fz | 9.5995 | 8.9193 | 8.6332 | O2 | 7.8049 | 7.1734 | 7.0697 |
| Fp2 | 9.3575 | 8.4117 | 9.1107 | O1 | 7.5260 | 6.9236 | 7.4002 |
| F8 | 9.3569 | 8.7205 | 8.4763 | F3 | 7.7258 | 7.0392 | 6.9440 |
| Cz | 9.2614 | 8.2537 | 8.7870 | F7 | 7.6354 | 6.9509 | 7.0236 |
| P3 | 7.2633 | 6.8134 | 6.7503 | T3 | 7.1219 | 6.7913 | 6.6266 |
| Pz | 7.0098 | 6.8035 | 6.6466 | | | | |

# F. Architectures and Training Details

This section gives a concise description of the components used in LERD and the training protocol.

## F.1. Input, preprocessing, and windowing

EEG is segmented into non–overlapping windows and $z$–scored channel–wise using statistics computed on the training split of each fold. In our AD experiments we use 2 s windows from 500 Hz recordings ($T=1000$) and $C=19$ electrodes (10–20 layout). Other datasets can adjust $C$ and $T$ without changing the architecture.

## F.2. Encoder

The encoder follows the temporal–spatial factorization popular in EEGNet, with a mild max–norm constraint on spatial depthwise kernels for stability on EEG.

- **Block 1 (temporal → spatial).** Depthwise temporal convolution → BatchNorm → ELU; then depthwise *spatial* convolution across electrodes → BatchNorm → ELU; time average pooling; dropout (0.1).

- **Block 2 (depthwise–separable temporal).** Depthwise temporal convolution → BatchNorm → ELU; pointwise mixing → BatchNorm → ELU; time average pooling; dropout (0.1).

- **Two branches.** (a) A flattened *main* feature vector is used by the classifier; (b) a temporally downsampled, per–electrode feature map feeds the EPDE/MELP/dLIF block.

## F.3. EPDE + MELP + dLIF coupling (latent event dynamics)

Given the encoder's per–electrode temporal features, the EPDE produces a differentiable posterior over next–event times. We parameterize a small MLP per channel to output mixture parameters for the **MELP** (lognormal mixture; $K=3$ components). Sampling is reparameterized during training; at test time we use mixture expectations. A compact hidden state is evolved with an explicit–Euler solver to obtain a denoised per–electrode trajectory used downstream. A rate proxy read from the EPDE is softly aligned with the **dLIF** prior via an $L_2$ rate consistency term with refractory gating and a plausible alpha/theta–to–beta frequency range.

## F.4. Event–Relational Graph (ERG) and GCN

From posterior samples of event times, cross–channel lags are mapped through a smooth STDP–shaped nonlinearity to edge scores in $[0, 1]$, then averaged over time/samples to produce a symmetric adjacency. A weak Fisher–$z$ alignment term biases edges toward observable correlations computed from the raw EEG without enforcing them. A single GCN layer converts per–channel temporal descriptors into compact node embeddings that are flattened for fusion.

## F.5. Classifier and fusion

We concatenate the encoder's main vector with the flattened GCN features and apply a two–layer MLP followed by a linear layer and Softmax over classes. No attention or recurrence is used at this stage; temporal information is already summarized

| Module | Input | Output | Activation |
|---|---|---|---|
| Input window | $\mathbb{R}^{B \times 1 \times 19 \times 1000}$ | – | – |
| Encoder (temporal branch) | $\mathbb{R}^{B \times 1 \times 19 \times 1000}$ | $\mathbb{R}^{B \times 19 \times L}$ ($L = 250$) | ELU |
| Encoder (main branch, flattened) | $\mathbb{R}^{B \times 1 \times 19 \times 1000}$ | $\mathbb{R}^{B \times 1178}$ | ELU |
| EPDE/MELP/dLIF block | $\mathbb{R}^{B \times 19 \times L}$ | $\mathbb{R}^{B \times 19 \times L}$ | ELU (MLPs), Tanh (ODE) |
| ERG adjacency | lags from EPDE/MELP | $\mathbb{R}^{B \times 19 \times 19}$ | exp kernel |
| GCN node embeddings | $\mathbb{R}^{B \times 19 \times L}$, adjacency | $\mathbb{R}^{B \times 19 \times 64}$ | ReLU |
| Flattened graph features | $\mathbb{R}^{B \times 19 \times 64}$ | $\mathbb{R}^{B \times 1216}$ | – |
| Fusion vector | concat(main, graph) | $\mathbb{R}^{B \times 2394}$ | – |
| Classifier logits | $\mathbb{R}^{B \times 2394}$ | $\mathbb{R}^{B \times |\mathcal{Y}|}$ | ReLU (hidden), Softmax (out) |

by EPDE/MELP and the encoder.

## F.6. Optimization and protocol

- **Loss.** Cross–entropy for labels plus small auxiliary terms: EPDE/MELP reconstruction/regularizers, dLIF rate consistency, ERG Fisher–$z$, and the IVP–KL surrogate. We use modest default weights and found them robust across cohorts.

- **Training.** Adam (lr $5 \times 10^{-4}$, weight decay $10^{-4}$), batch size 1024, gradient–norm clipping at 1.0, 30 epochs. Learning rate is halved if validation AUC does not improve for 15 epochs; the best AUC checkpoint is kept.

- **Evaluation.** Five–fold *cross–subject* splits; subject–level predictions from window probabilities via majority vote (or simple averaging).

## F.7. Shapes summary

Below we list only input/output sizes and activations for clarity; $L{=}250$ denotes the temporal length after the first time–pooling stage.

## F.8. Pseudocode summaries

---

**Algorithm 1** MELP Sampling (per electrode and time proxy)

---

**Require:** Mixture weights $w \in \Delta^{K-1}$, means $\mu \in \mathbb{R}^K$, standard deviations $\sigma \in \mathbb{R}_+^K$
**Ensure:** Inter-event interval $\tau \in \mathbb{R}_+$
 1: Sample component $k \sim \text{Categorical}(w)$
 2: Sample noise $\epsilon \sim \mathcal{N}(0,1)$
 3: $\tau \leftarrow \exp\big(\mu_k + \sigma_k \cdot \epsilon\big)$
 4: **return** $\tau$

---

**Algorithm 2** Neural ODE evolution over $[t_s, t_e]$ with $S$ Euler sub-steps

---

**Require:** Features $\xi$, projection map $\text{proj}(\cdot)$, decoder $\text{decode}(\cdot)$, vector field $f(\cdot)$, residual weight $\alpha > 0$, sub-steps $S \in \mathbb{N}$, interval $[t_s, t_e]$
**Ensure:** Decoded trajectory $\hat{z} \in \mathbb{R}^{\lfloor T/4 \rfloor}$
 1: $y_0 \leftarrow \text{proj}(\xi)$
 2: $\Delta t \leftarrow (t_e - t_s)/S$
 3: **for** $m = 0, 1, \ldots, S - 1$ **do**
 4: $\quad y_{m+1} \leftarrow y_m + \Delta t \cdot \big(f(y_m) + \alpha\, y_m\big)$
 5: **end for**
 6: $\hat{z} \leftarrow \text{decode}(y_S)$
 7: **return** $\hat{z}$

---

---

**Algorithm 3** Graph weights from $z_t$

---

**Require:** Per-electrode trajectories $z_t \in \mathbb{R}^{N \times L}$ with $L = \lfloor T/4 \rfloor$; scale $\gamma > 0$; kernel mode mode $\in$ {exp, gauss, inv1}
**Ensure:** Symmetric adjacency $W \in \mathbb{R}^{N \times N}$ with zero diagonal
1: $\forall c \in \{1, \ldots, N\}: s_c \leftarrow \frac{1}{L} \sum_{t=1}^{L} z_t(c, t)$
2: **for** $i = 1, \ldots, N$ **do**
3:     **for** $j = 1, \ldots, N$ **do**
4:         **if** $i = j$ **then**
5:             $W_{ij} \leftarrow 0$
6:         **else**
7:             $\Delta \leftarrow |s_i - s_j|$
8:             $W_{ij} \leftarrow \exp(-\gamma \, \Delta)$
9:         **end if**
10:     **end for**
11: **end for**
12: $W \leftarrow \frac{1}{2} (W + W^\top)$ {Enforce symmetry}
13: **return** $W$

---

---

**Algorithm 4** LERD: Training and Inference

---

**Require:** Dataset $\mathcal{D} = \{(X, Y)\}$, electrodes $C$, window length $T$, MELP comps $K$, ODE substeps $S$
**Require:** Loss weights $\lambda_{\text{aux}}, \lambda_{\text{spk}}, \lambda_{\text{graph}}, \lambda_{\text{LIF}}$; LR $\eta$
**Ensure:** Trained params $\Theta$; predictor LERD$(\cdot)$
1: Initialize encoder $\psi$, EPDE+MELP $\phi$, dLIF head $\xi$, ERG+GCN $\eta_g$, classifier $\theta$; Adam$(\eta)$
2: **for** epoch $= 1..E$ **do**
3:     **for** mini-batch $(X, Y) \sim \mathcal{D}$ **do**
4:         **Preprocess**: channel-wise $z$-score per window
5:         **Encoder** $(\psi)$: main_vec $\in \mathbb{R}^{B \times d_{\text{main}}}$, temp_feat $\in \mathbb{R}^{B \times C \times 1 \times L}$, $L = \lfloor T/4 \rfloor$
6:         **EPDE+MELP+ODE**:
      For $c = 1..C$: EPDE $\Rightarrow$ mixture $(\mathbf{w}_c, \boldsymbol{\mu}_c, \boldsymbol{\sigma}_c)$; sample $\tau = \exp(\mu_{c,k} + \sigma_{c,k}\varepsilon)$, accumulate events $\{T_c\}$; ODE evolve with $S$ Euler steps $\Rightarrow$ trajectory $z_{c,:}$
7:         Stack $Z \in \mathbb{R}^{B \times C \times L}$; keep event lags for ERG
8:         **dLIF prior**: compute $r_c(t)$ from $Z$ with refractory gate; rate proxy $\widehat{r}_c(t)$
9:         **Regularizers**: $\mathcal{R}_{\text{LIF}} = \sum_c \int (\widehat{r}_c - r_c)^2$; $\text{KL}_T = \sum_c G_c(-e^{-S})$ (finite-horizon IVP–KL regularizer)
10:         **ERG** from event lags: $e_{ij}(t) = \exp(-\alpha |\Delta \tilde{t}_{ij}(t)|)$, average $\Rightarrow \bar{A}$
11:         **Fisher–$z$ alignment**: $\mathcal{R}_{\text{ERG}} = \sum_{i<j} \left[ (z_{ij}^{\text{obs}} - z_{ij}^{\text{pred}})^2/(2\sigma_{ij}^2) + \frac{1}{2} \log \sigma_{ij}^2 \right]$
12:         **GCN** on $(\bar{A}, Z)$: $G \in \mathbb{R}^{B \times (C \cdot d_g)}$; **Fuse** $H = [\text{main\_vec}; G]$
13:         **Classifier**: logits $\ell = \text{MLP}(H)$, $p = \text{softmax}(\ell)$
14:         **Loss**: $\mathcal{L} = \text{CE}(p, Y) + \lambda_{\text{aux}} \mathcal{L}_{\text{STRODE}} + \lambda_{\text{spk}} \mathcal{L}_{\text{spk}} + \lambda_{\text{graph}} \mathcal{R}_{\text{ERG}} + \lambda_{\text{LIF}} \mathcal{R}_{\text{LIF}} + \text{KL}_T$
15:         Backprop; clip $\|\nabla\| \leq 1$; Adam step on $\Theta$
16:     **end for**
17: **end for**

18: **Inference on window** $X$:
19: Preprocess $\rightarrow$ Encoder; EPDE gives MELP expectations $\rightarrow$ events $\{T_c\}$; ODE $\rightarrow Z$; build $\bar{A}$; GCN $\rightarrow G$; fuse $\rightarrow H$; output $p$ and $\hat{y} = \arg\max p$
20: **Subject aggregation**: majority vote $\Rightarrow \hat{y}_{\text{subj}}$

---

