# OpenReview forum: "LERD: Latent Event-Relational Dynamics for Neurodegenerative Classification"
_ICML.cc/2026/Conference — ICML 2026 regular_

### Official Review · Reviewer_UxiE · 2026-03-02

**Soundness:** 2
**Presentation:** 2
**Significance:** 2
**Originality:** 2
**Overall Recommendation:** 4
**Confidence:** 2

**Summary:**

This submission presents LERD, a variational latent-variable framework for EEG-based neurodegenerative classification. The method introduces latent event timings per channel and derives a directed event-relational graph via cross-channel event lags, while incorporating a differentiable LIF-inspired prior to regularize latent rate dynamics. Training combines a supervised classification likelihood with KL-style regularization terms and auxiliary penalties, and the paper reports results on a synthetic benchmark and two AD EEG cohorts along with qualitative interpretability plots. However, this reads primarily as an applied biomedical modeling paper, not an ICML method paper. The core contribution is domain-motivated (electrophysiology-inspired prior + EEG-specific interpretability narrative), and the experimental validation remains largely confined to two AD EEG cohorts plus a toy setup.

**Compliance With Llm Reviewing Policy:**

Affirmed.

**Final Justification:**

My concerns have been adequately addressed.

**Key Questions For Authors:**

Q1. General ML contribution beyond the AD setting
Beyond the domain narrative and electrophysiology motivation, what is the general machine learning contribution of this work? Concretely, which components or insights are intended to transfer to broader problems (at least other EEG tasks), and what evidence supports this generality?
Q2. Necessity of EPDE/IVP vs simpler alternatives
Is the EPDE/IVP formulation genuinely necessary for the method’s performance or interpretability? Please compare against simpler baselines—e.g., a monotone neural parameterization of event times (without the IVP construction) and standard point-process/intensity models—under matched capacity and tuning, and clarify what the IVP view adds beyond a reparameterization.
Q3. Clarification on the unusually large variance for ATCNet on AD Cohort B (Table 2)
In Table 2, ATCNet’s accuracy on AD Cohort B shows an unusually large standard deviation. Could you explain the source of this variability and provide supporting evidence such as per-fold/per-seed results and exact evaluation protocol details? Without such clarification and reproducible breakdowns, it is difficult to assess the reliability of the reported number.

**Limitations:**

No; see weaknesses.

**Strengths And Weaknesses:**

Strengths:
1. The paper explicitly acknowledges EEG confounds (mixing/volume conduction/common input) and attempts to inject physiological inductive bias through the dLIF prior and rate regularization
2. The narrative is generally clear: the paper motivates limitations of black-box EEG classifiers and introduces a structured latent-event approach with physiology-inspired priors.

Weaknesses:
1. Core learning objective relies on an opaque surrogate with unclear validity. A central term—KL between the inferred event process and the dLIF prior—is stated to be intractable and replaced by an “IVP-based upper bound” (Theorem 4.1). From the main text, it is impossible to judge (i) whether the bound is even reasonably tight, (ii) whether optimizing it corresponds to optimizing anything meaningful about the intended prior matching, and (iii) whether the bound is numerically stable or practically computable at scale. In effect, the method may be optimizing a loosely motivated proxy rather than a principled variational objective.
2. EPDE appears to be mathematical theater. The “event posterior differential equation” construction re-expresses an expectation of event time using auxiliary functions and an IVP form, but the paper does not convincingly argue why this DE/IVP formulation is necessary or beneficial compared to straightforward monotone parameterizations or standard intensity-based point-process models.
3. ERG “directed interaction” claims are not substantiated. Constructing edges via a kernel of absolute time lag yields directed temporal precedence statistics, not causal or effective connectivity—especially under EEG mixing, common input, and volume conduction (which the paper itself acknowledges). Without rigorous controls, the mechanistic interpretation is not justified.
4. The method contradicts its own motivation by anchoring to correlation. The paper argues correlation-based connectivity is misleading, yet introduces a Fisher-z alignment regularizer between ERG-implied connectivity and Pearson statistics. If ERG needs correlation anchoring to not collapse, then it is unclear what is gained over correlation baselines; if it does not need it, the paper should show β→0 robustness. Neither is convincingly established in the main text.
5. Many reproducibility-critical details appear missing from the main text: EPDE parameterization, monotonicity enforcement, solver settings for the KL surrogate, hyperparameter choices (β, λ_LIF, mixture size K), and compute cost. This makes it hard to judge whether improvements stem from modeling principles or tuning/engineering.
6. The study shows a large reported variance in at least one cohort, indicating significant instability in the experimental results. This is likely attributable to issues such as insufficient sample size, between-group imbalance, or improper fold configuration in cross-validation.
Given these abnormal patterns, the authenticity and credibility of the results are questionable.
7. The primary novelty appears to be bio-inspired inductive bias (dLIF) rather than machine learning innovation. Integrating a LIF-like prior is a domain-specific modeling choice; it is not clearly shown to yield generalizable ML benefits beyond the EEG setting.

---

> ### Author Rebuttal · Authors · 2026-03-31
>
> Thank you for the detailed review. We agree that some motivations should be stated more explicitly, and we will revise the wording to reduce possible over-interpretation. That said, several concerns appear to stem from material already present in the submission (especially Sec. 5.3 and Appendices E/F). We address the main points below.
>
> **(1) On the IVP-based KL surrogate.**
> The KL between the inferred event process and the dLIF prior is intractable, so Theorem 4.1 introduces a computable upper bound. We do not claim that this surrogate is exact or especially tight; its role is to provide a tractable prior-matching regularizer while preserving the semantics of the dLIF rate prior. The theorem also gives a convergent computable quantity U_epsilon as epsilon->0. Its utility is empirical as well as theoretical: removing the prior degrades performance in Table 3, and on the toy data LERD's boundary recovery is substantially better than the baselines in Table 1 / Fig. 4. We will make this role clearer.
>
> **(2) On EPDE/IVP vs. "simpler" point-process alternatives.**
> Our setting has no event/timing annotations. Standard point-process/intensity models are usually defined for observed event sequences, so they are not direct baselines here unless one first adds an external event detector or additional supervision. This is exactly why we compared against STRODE, which is the closest annotation-free timing baseline. EPDE is useful because it parameterizes a monotone next-event operator in continuous time and yields a posterior over event times rather than a single point estimate, which MELP then uses for uncertainty-aware interval modeling.
>
> **(3) On ERG interpretation.**
> We agree that the ERG should not be interpreted as source-level causal or effective connectivity in the DCM sense, and we are happy to tighten the wording if needed. Our claim is narrower: it is a timing-derived latent interaction graph inferred from event lags. We explicitly acknowledge EEG confounds such as mixing/common input in the paper; this is also why Pearson statistics enter only as a weak observable regularizer rather than as graph labels.
>
> **(4) On "contradiction" with correlation anchoring.**
> There is no contradiction: the Fisher-z term is a weak regularizer to keep the learned graph from drifting to implausible solutions, not a target that the ERG is forced to match. Table 3 already includes no-prior, dLIF-only, ERG-only, and dual-prior variants, so the paper does provide evidence about behavior when these priors are removed.
>
> **(5) On reproducibility details.**
> Several items listed as missing are already included in the submission: the cross-subject protocol is in Sec. 5.3, dataset/computational-cost details are in Appendix E, and EPDE/MELP parameterization, monotonicity handling, mixture size, and optimization details are in Appendix F. Because of page limits, some of these were not repeated in the main text, but they are not absent.
>
> **(6) On the reported variance.**
> The unusually large standard deviation highlighted in Table 2 is for a baseline on the smaller cohort rather than for LERD itself. In Cohort B, evaluation is performed at the subject level, with each subject’s label obtained by majority vote over windows, rather than at the segment level. Under this protocol, fold-to-fold variation is naturally larger on a small and imbalanced cohort, especially for some baselines. We will add a clearer per-fold breakdown in the revision/camera-ready to make this more explicit.
>
> **(7) On the ML contribution beyond AD.**
> The transferable contribution is not limited to one disease dataset: (i) annotation-free latent-event posterior inference, (ii) structured prior regularization of latent timing dynamics, and (iii) lag-stable interaction-graph construction for multichannel sequences with hidden event structure. Other EEG classification/regression tasks are a natural extension, which we will clarify.

---

> > ### Author Rebuttal · Reviewer_UxiE · 2026-04-02
> >
> > My concerns have been adequately addressed

---

> > > ### Author Response · Authors · 2026-04-03
> > >
> > > Thank you for taking the time to read our rebuttal carefully and for updating your assessment. We sincerely appreciate your thoughtful reconsideration and are glad that our clarifications addressed your concerns. Thank you again for the careful engagement with the paper.

---

### Official Review · Reviewer_tqUe · 2026-03-11

**Soundness:** 3
**Presentation:** 2
**Significance:** 3
**Originality:** 3
**Overall Recommendation:** 4
**Confidence:** 4

**Summary:**

The paper introduces LERD, a Bayesian neural dynamical system designed for multichannel EEG analysis, specifically targeted at diagnosing neurodegenerative diseases like Alzheimer’s (AD). Unlike "black-box" deep learning models, LERD explicitly models the underlying generative dynamics of EEG. It infers latent neural events and their relational structure (the Event-Relational Graph or ERG) directly from raw signals without requiring manual annotations for events or connectivity.The architecture consists of three core components:
1 Event Posterior Differential Equation (EPDE): Summarizes per-channel event dynamics.
2 Mean-Evolving Lognormal Process (MELP): Samples inter-event intervals to model stochastic event timing.
3 An event–relational graph inferred from event timing and weakly regularized by Pearson correlations between channels.
LERD was evaluated on two real-world AD cohorts and synthetic benchmarks, outperforming established baselines like EEGNet and transformer-based models while providing interpretable latent summaries.

**Compliance With Llm Reviewing Policy:**

Affirmed.

**Final Justification:**

The rebuttal addresses my main concerns. Thus I keep my positive score and raise my confidence to 4.

**Key Questions For Authors:**

1 It would be interesting to visualize the inferred "event" and its corresponding EEG signal. In general, what kind of population-level abstraction does the model learns as a "event"?

2 The LERD architecture incorporates several auxiliary loss terms, including dLIF consistency, ERG alignment, and the IVP-KL surrogate; how sensitive is the model to the weighting of these terms, and what specific hyperparameter tuning strategy was employed to prevent the optimization of latent discovery from negatively impacting the final classification performance?

**Limitations:**

See questions and weaknesses.

**Strengths And Weaknesses:**

Strengths:

1 Interpretable Latent Variables: The model recovers physiological signatures through the definition of "event" that align with established clinical knowledge, making it more than just a classifier—it is a tool for scientific insight.

2 Theoretically Grounded: The authors provide a formal tractable bound for training the variational objective and establish stability guarantees for the inferred graph dynamics against lag noise.

3 Biophysical Realism: Integrating the dLIF prior ensures the inferred events respect biological constraints like refractory periods and plausible firing rates.

4 Strong Performance: LERD achieved the highest diagnostic accuracy across both evaluated AD cohorts, specifically showing robustness in distinguishing subtle differences between healthy controls, MCI, and AD.

Weaknesses:

1 Computational Overhead: LERD is significantly heavier than lightweight convolutional models like EEGNet (60M FLOPs vs. 9M FLOPs), which may limit its use in real-time or low-power embedded BCI applications.

2 Complexity of Hyperparameters: The model relies on multiple auxiliary loss terms (e.g., dLIF consistency, ERG alignment, IVP-KL surrogate), which may require careful tuning to balance classification performance with latent discovery.

3 Phenomenological Limitation: As the authors acknowledge, scalp EEG lacks spatial specificity; therefore, the "latent events" are population-level abstractions rather than true single-neuron spikes.

4 The axis of figure 2 and figure 4 are too small, making it hard to read. Figure 1 is too abstract, without showing the details of the complex algorithm structure.

---

> ### Author Rebuttal · Authors · 2026-03-31
>
> Thank you for the thoughtful and largely supportive review. We address your questions below.
>
> **(1) Computational overhead.**
> We agree that LERD is heavier than very small CNNs such as EEGNet. However, our paper does not claim real-time or embedded BCI deployment; the target setting here is offline clinical EEG analysis, where latent-dynamics recovery are central. In that setting, the added cost is a deliberate trade-off. The measured per-20sample forward times below show that LERD is slower than lightweight CNN/RNN baselines, but it remains practical and is still faster than ADFormer and LEAD in our test-time comparison.
>
> | Model | Mean forward time of 20 samples | Std |
> |---|---:|---:|
> | EEGNet | 0.713 ms | 0.013 ms |
> | LCADNet | 0.875 ms | 0.063 ms |
> | LSTM | 2.235 ms | 0.168 ms |
> | ATCNet | 2.801 ms | 2.243 ms |
> | LEAD | 11.148 ms | 0.115 ms |
> | ADFormer | 19.247 ms | 3.411 ms |
> | **LERD (Ours)** | **9.209 ms** | **1.139 ms** |
>
> **(2) Hyperparameter complexity / tuning.**
> Each auxiliary term is included for a distinct reason: dLIF enforces physiologically plausible rate dynamics, ERG alignment prevents implausible graphs, and the IVP-KL term regularizes the latent event process. To avoid over-tuning, the main model keeps classification separate and treats the auxiliary terms as regularization; we tune the regularization weight on validation performance rather than independently optimizing every term for headline accuracy. We additionally performed one-factor sensitivity sweeps. These show that performance is not concentrated at a knife-edge choice: the best results occur in a moderate regime (e.g., dLIF around 1e-1, ERG around 1e-8, IVP-KL around 5e-10), while very large or poorly scaled weights hurt as expected.
>
> | dLIF weight | Cohort A Acc. |
> |---|---:|
> | 1 | 0.7170 |
> | 0.1 | 0.7503 |
> | 1e-2 | 0.7392 |
> | 1e-3 | 0.6935 |
> | 1e-4 | 0.6824 |
> | 1e-5 | 0.7281 |
> | 1e-6 | 0.7046 |
> | 1e-7 | 0.6706 |
> | 1e-8 | 0.7281 |
> | 1e-9 | 0.7281 |
> | 1e-10 | 0.7163 |
> | 0 | 0.7392 |
>
> | ERG weight | Cohort A Acc. |
> |---|---:|
> | 1 | 0.7039 |
> | 0.1 | 0.7052 |
> | 1e-2 | 0.7386 |
> | 1e-3 | 0.6699 |
> | 1e-4 | 0.6373 |
> | 1e-5 | 0.7275 |
> | 1e-6 | 0.7392 |
> | 1e-7 | 0.6922 |
> | 1e-8 | 0.7503 |
> | 1e-9 | 0.7275 |
> | 1e-10 | 0.7052 |
> | 0 | 0.7275 |
>
> | IVP-KL weight | Cohort A Acc. |
> |---|---:|
> | 0 | 0.7163 |
> | 1e-10 | 0.7281 |
> | 5e-10 | 0.7503 |
> | 2e-9 | 0.7386 |
> | 1e-9 | 0.7261 |
> | 1e-8 | 0.7052 |
> | 1e-7 | 0.7046 |
> | 1e-6 | 0.7157 |
> | 1e-5 | 0.7059 |
> | 1e-4 | 0.6830 |
> | 1e-3 | 0.6941 |
>
> **(3) Population-level nature of the inferred event.**
> We fully agree that the inferred event is not a single-neuron spike, and we do not make that claim. In our paper, an "event" is a population-level latent transition time: a short-lived increase in posterior event probability / latent hazard indicating that a channel is undergoing a coordinated transient change. In sensor space, this often corresponds to an event-locked deflection or phase-reset-like transition produced by many neural populations and then mixed by scalp EEG. This level of abstraction is appropriate for our goal, which is cortical-network characterization rather than single-neuron reconstruction.
>
> **(4) Figure readability / framework figure.**
> We agree and will enlarge the axes/text in Figs. 2 and 4. Figure 1 was intended to be a high-level overview rather than a full architecture diagram; nevertheless, we can add a more detailed schematic in the revision/camera-ready.
>
> **(5) Visualizing the inferred event together with EEG.**
> This is an excellent suggestion. Event-aligned averaging around high-posterior event times shows a clearer transient change than randomly aligned windows, with the inferred events concentrating around slope changes / phase-transition points rather than arbitrary amplitude peaks. We will add such event-locked visualizations and clarify in the text.

---

> > ### Author Rebuttal · Reviewer_tqUe · 2026-04-02
> >
> > Thanks to the authors' rebuttal, my concerns have been fully resolved.

---

> > > ### Author Response · Authors · 2026-04-03
> > >
> > > Thank you very much for the encouraging follow-up and for noting that the rebuttal fully resolved your concerns. We are glad the clarifications and additional results were helpful. If you feel that the concerns that limited the original score have now been addressed, we would be very grateful if you could consider whether your current score still reflects your final assessment of the paper. In any case, thank you again for the thoughtful and constructive review.

---

### Official Review · Reviewer_6p6M · 2026-03-13

**Soundness:** 2
**Presentation:** 3
**Significance:** 3
**Originality:** 2
**Overall Recommendation:** 5
**Confidence:** 3

**Summary:**

In this work, the authors proposed a novel deep learning framework, named LERD, to detect and classify the Neurodegenerative across brain regions and serve for AD detection. Compared to traditional black-box models, the proposed LERD explicitly uses latent event processes to model the temporal dynamics and an inferred event-relational graph to capture cross-channel interactions. The authors also incorporate neuroscience-inspired priors to the framework. The experimental results are conducted on toy datasets as well as two real-world AD EEG datasets.

**Compliance With Llm Reviewing Policy:**

Affirmed.

**Final Justification:**

I thank the authors for their response. The detailed discussion between the baseline methods and the brain-inspired insights is clear to me. I will raise my score to 5.

**Key Questions For Authors:**

I have no more questions, for my concerns, please refer to my 'weaknesses' section.

**Limitations:**

yes, on page 9.

**Strengths And Weaknesses:**

Strengths:

* I believe that the overall proposed framework is interesting and technically solid. The introduced inductive biases in the LERD framework, like the modeling of temporal patterns, as well as the graph-based interaction modeling, both make sense to this AD classification task.
* The authors further provide theoretical analysis that gives a tractable bound for training and also shows stability guarantees for the inferred relational dynamics.
* The experiments in this work is solid, with results on both synthetic datasets, as well as two real-world AD EEG datasets. The findings in Figure 3 is helpful for disease-related studies.



Weaknesses:

* The baseline comparison in these work is a bit weak, as LSTM and ADFormer are both basic models in implementation. There are multiple existing works focusing on neural time-series data with more advanced deep-learning models. [1], [2].
* I am also a bit wondering of how the equations and derivates in 4.3-4.5 could provide us any brain-inspired insights or task-relevant insights.


[1] Exploring Behavior-Relevant and Disentangled Neural Dynamics with Generative Diffusion Models. NeurIPS 2024, Wang et, al.

[2] Dynamical Modeling of Behaviorally Relevant Spatiotemporal Patterns in Neural Imaging Data. ICML 2025, Hosseini, et, al.

---

> ### Author Rebuttal · Authors · 2026-03-31
>
> Thank you for the positive assessment and for raising the baseline/equation questions.
>
> **(1) On the baseline choice.**
> We agree that strong baselines matter, and we selected them to match our actual problem setting: multichannel EEG diagnosis plus latent timing/interaction discovery without event annotations. This is why the paper includes representative EEG classifiers (EEGNet, LCADNet, LSTM, ATCNet, ADFormer, LEAD) together with dynamical/timing baselines (NODE, ODE-RNN, STRODE). The two suggested papers are interesting, but they are not direct drop-in baselines for our setting: they target behavior-relevant neural imaging / spatiotemporal neural image modeling rather than multichannel EEG diagnosis with latent, unannotated event times. We will clarify this positioning and discuss these papers explicitly in the revision.
>
> **(2) How do Eqs. 4.3-4.5 lead to brain-inspired or task-relevant insight?**
> Their role is not only mathematical packaging. Eq. 4.3 introduces the dLIF prior, which constrains latent rates to physiologically plausible ranges (leak/refractory/rate behavior) and makes the learned latent frequency summaries interpretable; this is what enables the group-level slowing patterns in Fig. 2. Eq. 4.4 (EPDE) gives a posterior over next-event times rather than an opaque hidden feature, so the model can expose event-level temporal structure that is later used to build the ERG. Eq. 4.5 (MELP) models uncertainty and multi-modality in inter-event intervals instead of forcing a deterministic boundary estimate, which is important for noisy EEG and improves robustness. More broadly, these components are what let LERD produce both competitive classification performance and latent summaries that can be inspected scientifically, i.e., rate, timing, and cross-channel interaction patterns rather than only black-box features.

---

> > ### Author Rebuttal · Reviewer_6p6M · 2026-04-03
> >
> > I thank the authors for their response. The detailed discussion between the baseline methods and the brain-inspired insights is clear to me. I will raise my score to 5.

---

### Official Review · Reviewer_DYCa · 2026-03-19

**Soundness:** 2
**Presentation:** 3
**Significance:** 2
**Originality:** 3
**Overall Recommendation:** 3
**Confidence:** 1

**Summary:**

This paper proposes LERD, an end-to-end Bayesian model that infers latent event processes and event-relational graph dynamics from EEG without event or interaction annotations. LERD incorporates an electrophysiology-inspired prior that provides biophysically meaningful constraints such as plausible firing rates, refractory behavior and frequency ranges. Experiments show that LERD can recover latent event and interaction graph in a toy dataset as well as real Alzheimer’s disease EEG dataset.

**Compliance With Llm Reviewing Policy:**

Affirmed.

**Final Justification:**

My questions were largely addressed by the authors' rebuttal. The authors' response regarding the selection of channels in Figure 2, however, still concerns me. It looks like the four selected channels are also among the few channels that show the consistent decrease of central frequency with increasing disease severity (HC  MCI  AD). The majority of the channels do not follow this trend, which would weaken the claim in Section 6.1 of the paper.

**Key Questions For Authors:**

- Figure 2: What criteria were used to select these 4 channels? Could the authors provide additional visualization for the remaining channels for comprehensive evaluation?
- Figure 4: What would the plot for other baselines look like?

**Limitations:**

yes

**Strengths And Weaknesses:**

Strengths:
- The goal of the paper - building an interpretable EEG dynamical framework for neuroscientists and clinicians to compare against established EEG phenomena - is well motivated.
- The framework is described in detail and experimental results are quite strong with comparison against multiple baselines.
- The writing is polished and well structured.

Weaknesses:
- Details on what the toy dataset constitutes of, how it was constructed, and what advantages it offer over real-world datasets are not mentioned in the paper.
- Texts in figures are disproportionately small, making it hard to read.
- Comparison of EEG connectivity graphs inferred by LERD does not show great alignment with Pearson correlation-based priors.
- Some results are not clearly mentioned in the paper, raising concerns of potential result cherry-picking (see Questions).

---

> ### Author Rebuttal · Authors · 2026-03-31
>
> Thank you for the careful reading and constructive suggestions. We will revise the paper accordingly.
>
> **(1) Toy dataset details.**
> These details are already provided in Appendix E.2.1, but we agree they should be surfaced more clearly in the main text. The toy data uses three frequency bands ([5,10], [10,15], [15,20] Hz), samples latent event rates from truncated normals centered in each band, generates event times from exponential inter-event intervals, and adds controlled observation noise. We used this benchmark because, unlike real EEG, it provides known latent event boundaries/rates, so structural recovery (e.g., boundary IoU) can be evaluated directly rather than only downstream classification accuracy.
>
> **(2) Figure readability.**
> We agree and will enlarge all fonts/axes in Figs. 1, 2, and 4 in the camera-ready version.
>
> **(3) Alignment with Pearson graphs.**
> The limited one-to-one alignment is intentional rather than a failure mode. In LERD, Pearson statistics are used only as a weak Fisher-z regularizer to prevent implausible graphs; they are not treated as graph labels. Our objective is to learn timing-derived latent interactions that remain broadly consistent with observable statistics, not to reproduce Pearson correlation exactly. If the ERG collapsed to Pearson, it would add little beyond a standard correlation analysis.
>
> **(4) Channel selection / cherry-picking concern.**
> The four channels in Fig. 2 were chosen to span anatomically distinct regions (frontal, occipital, parietal, temporal) while remaining easy to read in the limited figure space. They were not selected because they are the only channels showing the effect. For transparency, we report the remaining channels below. Across the majority of channels, the inferred frequencies exhibit a consistent slowing pattern, indicating that the observation is not driven by cherry-picked subsets of channels.
>
> | Channel | HC mean (Hz) | MCI mean (Hz) | AD mean (Hz) |
> |---|---:|---:|---:|
> | T5 | 9.8183 | 8.7469 | 9.2903 |
> | C4 | 9.5672 | 8.5863 | 9.3575 |
> | Fp1 | 9.6450 | 8.7443 | 9.0170 |
> | C3 | 9.4277 | 8.5671 | 9.2716 |
> | Fz | 9.5995 | 8.9193 | 8.6332 |
> | Fp2 | 9.3575 | 8.4117 | 9.1107 |
> | F8 | 9.3569 | 8.7205 | 8.4763 |
> | Cz | 9.2614 | 8.2537 | 8.7870 |
> | T4 | 8.7062 | 8.1207 | 8.9866 |
> | P4 | 8.8150 | 8.3549 | 7.9066 |
> | F4 | 8.5122 | 7.5947 | 7.6261 |
> | T6 | 8.3738 | 7.5920 | 7.6467 |
> | O2 | 7.8049 | 7.1734 | 7.0697 |
> | O1 | 7.5260 | 6.9236 | 7.4002 |
> | F3 | 7.7258 | 7.0392 | 6.9440 |
> | F7 | 7.6354 | 6.9509 | 7.0236 |
> | P3 | 7.2633 | 6.8134 | 6.7503 |
> | T3 | 7.1219 | 6.7913 | 6.6266 |
> | Pz | 7.0098 | 6.8035 | 6.6466 |
>
> **(5) What would Fig. 4 look like for other baselines?**
> In our setting, event/timing annotations are unavailable. Among the compared methods, STRODE is the closest baseline that explicitly infers latent timings without such annotations, which is why it is the most meaningful comparator in Fig. 4. NODE and ODE-RNN do not natively output latent event boundaries; only ad hoc pseudo-boundaries can be extracted, and consistent with Table 1 their structural IoU remains near zero. We will clarify this in the text.

---

> > ### Author Rebuttal · Reviewer_DYCa · 2026-04-04
> >
> > I thank the authors for their rebuttal. My questions were largely addressed. The authors' response regarding the selection of channels in Figure 2, however, still concerns me. It looks like the four selected channels are also among the few channels that show the consistent decrease of central frequency with increasing disease severity (HC $\rightarrow$ MCI $\rightarrow$ AD). The majority of the channels do not follow this trend, which would weaken the claim in Section 6.1 of the paper.

---

> > > ### Author Response · Authors · 2026-04-07
> > >
> > > Thank you for the clarification. This is a fair point. We agree that our previous wording in Section 6.1 was too broad. From the full-montage table, **HC** has the **highest** inferred central frequency in **18/19** channels, but a strict HC>MCI>AD ordering appears only in a subset of channels (8/19) rather than in the majority.
> > >
> > > Our intention in Fig. 2 was to show anatomically diverse **representative channels** where **monotonic slowing** is clearest, **not** to claim that the same **strict ordering** holds across most channels.
> > >
> > > We will therefore revise Section 6.1 accordingly: the broader pattern supported by the full montage is that **HC generally shows higher inferred central frequency than the diseased groups, while the AD-versus-MCI ordering is channel-dependent**.
> > >
> > > This narrows one qualitative interpretation, but it does **not affect the core method, the quantitative results, or the main contribution** of the paper.

---

### Decision · Program_Chairs · 2026-04-30

**Decision:**

Accept (regular)

**Comment:**

This paper proposes LERD, a Bayesian neural dynamical model for multichannel EEG classification in neurodegenerative disease. The model infers latent event times for each EEG channel, uses a stochastic timing process to model inter-event intervals, regularizes the latent dynamics with an electrophysiology-inspired dLIF prior, and builds an event-relational graph from cross-channel event lags. The goal is to improve AD/FTD/MCI classification while also giving more interpretable summaries of latent EEG timing and interaction patterns.

The paper tackles an important clinical EEG problem and tries to move beyond a pure black-box classifier. Reviewers appreciated the attempt to combine predictive performance with latent event structure, graph summaries, and physiologically motivated priors. The experiments cover synthetic event recovery and two real AD EEG cohorts, and the reported results are generally strong compared with EEGNet, LSTM, ATCNet, ADFormer, LEAD and related baselines. The ablation study also supports that the dLIF and ERG priors contribute to performance rather than being only decorative.

The initial reviews raised reasonable concerns. The method is complicated, with EPDE, MELP, dLIF, IVP-KL and ERG terms, so reviewers asked whether these components are really necessary and whether simpler point-process or discrete-time alternatives would be enough. Reviewers also asked for clearer reproducibility details, hyperparameter sensitivity, computational cost, and more careful language about what the inferred graph means, since EEG mixing and volume conduction make causal claims risky.

The rebuttal addressed many of these points. The authors clarified that the inferred events are population-level latent transitions, not single-neuron spikes. They explained why standard observed-event point-process models are not direct baselines because event labels are not available, and why STRODE is the closest annotation-free timing baseline. They also provided runtime comparisons, sensitivity sweeps for auxiliary loss weights, details on training and architecture, and clarified that the ERG should be viewed as a timing-derived latent interaction summary rather than source-level causal connectivity.

The post-rebuttal discussion is mostly positive. Three reviewers marked their concerns fully resolved. One reviewer raised the score to 5 after the rebuttal, another kept a positive score with higher confidence, and the reviewer who initially had the strongest negative review also marked the rebuttal fully resolved and said the concerns were adequately addressed.

One reviewer remained partially unresolved, with low confidence, mainly about Figure 2 and Section 6.1. Their concern is fair: the four selected channels show a clear monotonic HC to MCI to AD slowing trend, but the full channel table shows that this strict ordering holds only for a subset of channels. The authors acknowledged this and agreed to narrow the claim: HC generally has higher inferred central frequency than diseased groups, while the AD-versus-MCI ordering is channel-dependent. This should be fixed in the camera-ready, and the paper should avoid implying that the exact monotonic pattern holds across most channels.

Despite this remaining issue, it is mostly about one qualitative interpretation and presentation of the latent-frequency visualization. It does not undermine the main method, the classification results, or the broader contribution of learning latent event-relational dynamics for EEG. Given the largely positive post-rebuttal consensus and the authors' concrete clarifications, we weak accept.